# Early-life social experience affects offspring DNA methylation and later life stress phenotype

Zachary M. Laubach [1,2,3,4,13✉], Julia R. Greenberg[1,2,4], Julie W. Turner [1,2,3,4], Tracy M. Montgomery[1,2,4,5], Malit O. Pioon[4], Maggie A. Sawdy[1,2], Laura Smale[1,6], Raymond G. Cavalcante [7], Karthik R. Padmanabhan[7], Claudia Lalancette[7], Bridgett vonHoldt [8], Christopher D. Faulk [9], Dana C. Dolinoy[7,10,11], Kay E. Holekamp [1,2,3,4] & Wei Perng[12]

Studies in rodents and captive primates suggest that the early-life social environment affects future phenotype, potentially through alterations to DNA methylation. Little is known of these associations in wild animals. In a wild population of spotted hyenas, we test the hypothesis that maternal care during the first year of life and social connectedness during two periods of early development leads to differences in DNA methylation and fecal glucocorticoid metabolites (fGCMs) later in life. Here we report that although maternal care and social connectedness during the den-dependent life stage are not associated with fGCMs, greater social connectedness during the subadult den-independent life stage is associated with lower adult fGCMs. Additionally, more maternal care and social connectedness after den independence correspond with higher global (%CCGG) DNA methylation. We also note differential DNA methylation near 5 genes involved in inflammation, immune response, and aging that may link maternal care with stress phenotype.

[1] Department of Integrative Biology, Michigan State University, East Lansing, MI, USA. [2] Program in Ecology, Evolution, and Behavior, Michigan State University, East Lansing, USA MI, USA. [3] BEACON, NSF Center for the Study of Evolution in Action, Michigan State University, East Lansing, MI, USA. [4] Mara Hyena Project, Masai Mara National Reserve, Narok, Kenya. [5] Max Planck Institute of Animal Behavior, Department for the Ecology of Animal Societies, Konstanz, Germany. [6] Department of Psychology, Michigan State University, East Lansing, MI, USA. [7] Epigenomics Core, University of Michigan, Ann Arbor, MI, USA. [8] Department of Ecology and Evolutionary Biology, Princeton University, Princeton, NJ, USA. [9] Department of Animal Sciences, University of Minnesota, St. Paul, MN, USA. [10] Department of Environmental Health Sciences, University of Michigan School of Public Health, Ann Arbor, MI, USA. [11] Department of Nutritional Sciences, University of Michigan School of Public Health, Ann Arbor, MI, USA. [12] Department of Epidemiology and Lifecourse Epidemiology of Adiposity and Diabetes (LEAD) Center, University of Colorado Denver, Aurora, CO, USA. [13] Present address: Department of Ecology and Evolutionary Biology, University of Colorado, Boulder, CO, USA. ✉email: zachary.laubach@colorado.edu

Early social experiences shape many aspects of an organism's future phenotype. Over 60 years ago, experiments with laboratory rodents and non-human primates revealed that maternal care and interactions with peers have lasting effects on an organism's stress physiology and behavior[1,2]. In rodents, lower rates of maternal licking and grooming during the offspring's first 10 days of life caused elevated adult plasma corticosterone in response to external stressors[3] and exhibition of fearful behaviors[4,5]. Beyond the beneficial effect of maternal care, interactions with group members were protective against an elevated corticosterone response to a standardized stressor in Sprague–Dawley rats[6]. Similarly, the physiological toll of maternal separation during early life later manifested as a flat cortisol trajectory in rhesus macaques[7]. The importance of early social experiences extends to our own species. For example, children who lived in orphanages had altered hypothalamic-pituitary-adrenal (HPA) activity, including elevated basal cortisol levels that remained evident even after adoption[8]. This body of literature emphasizes the importance of maternal care and early social interactions in the ontogenetic development of stress phenotypes. Yet, the underlying biological mechanisms remain unclear.

DNA methylation is a mitotically stable epigenetic mark that is responsive to environmental cues and is involved in the regulation of gene expression[9]. It has been championed as a potential biological mechanism linking early social experiences to later stress phenotype. A landmark cross-fostering study in rodents showed that lower rates of maternal licking and grooming corresponded with higher DNA methylation of the promoter region of the hippocampal glucocorticoid receptor (GR; *NR3C1*) gene, lower GR RNA expression, and elevated plasma corticosterone among adult offspring[10]. Subsequent work in rodents demonstrated that maternal separation is correlated with widespread differences in brain tissue DNA methylation, not only at specific promoter regions but also in the form of lower global DNA methylation[11]. In rhesus macaques, maternal vs. peer-led rearing corresponded with genome-wide changes in DNA methylation in brain tissue and T cells during adulthood[12]. Finally, human epidemiological studies show that lower quality of maternal care corresponds with higher blood leukocyte DNA methylation of brain-derived neurotrophic factor (*BDNF*) and oxytocin receptor (*OXTR*) in adulthood[13].

Although there is a large body of literature on the topic of early social experience as a determinant of an organism's future stress phenotype and DNA methylation as a potential mediator, we identify three major lacunae. First, there is a lack of studies that measure natural variation in the quantity or quality of early social experience, as much of this research includes experimental studies involving maternal separation and peer isolation, which are routine procedures in studies of captive primates and rodents[14,15]. While informative, experiences of extreme deprivation do not capture the range of normative social interactions that are relevant to development[3,16,17]. Second, although studies have examined the relationships among social experiences, DNA methylation, and stress phenotype in a piecemeal fashion, few have measured all three components in the same population[10,18–20]. Doing so is necessary to explicitly test the hypothesis that DNA methylation represents a mechanistic link between early social experience and future stress phenotype. Third, few studies explicitly consider that variation in the type and/or timing of social experiences may affect DNA methylation[21] and stress phenotypes[22], particularly in wild animal populations where development is subject to environmental fluctuation and pressures of natural selection.

We address these three gaps in the literature by examining the relationships among early social experience, DNA methylation, and stress phenotype in a long-term field study of wild spotted hyenas (*Crocuta crocuta*). Based on earlier work in primates and rodents, our primary hypothesis is that unfavorable early social experiences correspond with adverse stress physiology later in life, as indicated by higher fecal glucocorticoid concentrations. Second, we hypothesize that these early social experiences influence patterns of DNA methylation measured in subadult and adult hyenas. Third, we hypothesize that differential DNA methylation is on the causal pathway, and thus a potential mechanism linking early social experience to future stress physiology.

Here, we show that maternal care and social connectedness during the earlier den-dependent life stage are not associated with fecal glucocorticoid metabolites (fGCMs; an indicator of stress phenotype). However, greater social connectedness during the later subadult den-independent life stage is associated with lower adult fGCMs. In addition, more maternal care and social connectedness after den independence corresponds with higher global (%CCGG) DNA methylation, a marker of genomic stability and overall health. We also note differential DNA methylation near five genes involved in inflammation, immune response, and aging that may link maternal care with stress phenotype. Our findings suggest that both maternal care during the first year of life and social connections after leaving the den influence DNA methylation and contribute to a developmentally plastic stress response.

## Results

Study population characteristics and results from bivariate analyses are provided in the Supplementary Material.

**Part 1: associations among early social environment and adult fGCMs.** Among hyena cubs, none of the maternal care or social connectivity measures during the communal den (CD) period were associated with adult fGCMs levels in the adjusted models (Table 1). However, during the later den-independent (DI) period of development, higher network degree, and strength were each associated with lower adult fGCMs after adjusting for a hyena's sex, level of human disturbance in its birth year, age, reproductive state, and the time of day of sample collection (−11% [95% confidence interval (CI): −20, −2] and −13% [95% CI: −23, −3] for network degree and strength, respectively; Table 1).

**Part 2: associations between early social environment and global DNA methylation.** We observed multiple positive associations between early social experiences and %CCGG methylation that were robust to covariate adjustment. In our demographic covariates model, which adjusted for offspring sex and age, we found that each 1-SD greater proportion of time spent in close proximity to the mother corresponded with 1.36 (95% CI: 0.72, 2.03) greater %CCGG methylation in cub and subadult offspring. The association was robust to adjustment in the social experience covariates model (1.28 [95% CI: 0.34, 2.20] %CCGG), and the ecological covariates model (1.07 [95% CI: 0.34, 1.83] %CCGG). In the model that included all covariates, there was a slight attenuation of the estimate, and the confidence interval narrowly crossed the null (0.90 [95% CI: −0.04, 1.86] %CCGG; Fig. 1). We also found that more time spent nursing was associated with higher %CCGG methylation in models that accounted for the demographic covariates (0.86 [95% CI: 0.14, 1.60] %CCGG) and social experience (1.12 [95% CI: 0.10, 2.08] %CCGG) covariates. However, adjustment for ecological covariates (human disturbance and migration status in the birth year) attenuated the effect of nursing time on %CCGG methylation. There was no effect of proportion of time spent grooming on %CCGG methylation (Fig. 1), and maternal care and maternal rank interactions were not significant.

**Table 1 Association of maternal care and early-life social network metrics with adult fGCMs.**

| | $\beta$ (95% CI) fGCMs[a] | | | |
|---|---|---|---|---|
| | **N** | **Unadjusted model** | **N** | **Adjusted model[b]** |
| Maternal care FAS (per 1-SD) | | | | |
| Close proximity | 74 | −0.13 (−0.28, 0.01) | 73 | −0.01 (−0.16, 0.13) |
| Nursing | 74 | −0.10 (−0.26, 0.06) | 73 | 0.04 (−0.11, 0.18) |
| Grooming | 74 | −0.06 (−0.21, 0.08) | 73 | 0.00 (−0.13, 0.13) |
| CD period (per 1-SD) | | | | |
| Degree | 79 | −0.13 (−0.26, −0.02) | 79 | −0.07 (−0.19, 0.04) |
| Strength | 79 | −0.11 (−0.21, 0.01) | 79 | −0.05 (−0.15, 0.06) |
| Betweenness | 79 | −0.03 (−0.19, 0.12) | 79 | −0.04 (−0.18, 0.10) |
| DI period (per 1-SD) | | | | |
| Degree | 79 | −0.19 (−0.30, −0.08) | 79 | −0.12 (−0.22, −0.02) |
| Strength | 79 | −0.16 (−0.29, −0.04) | 79 | −0.14 (−0.26, −0.03) |
| Betweenness | 79 | −0.10 (−0.26, 0.07) | 79 | −0.04 (−0.18, 0.11) |

Confidence intervals based on percentile bootstrapping; 2000 simulations.
fGCMs fecal glucocorticoid metabolites.
[a] Beta estimates are fGCM concentrations (ng/g) on the natural log scale from mixed models in which hyena ID was included as a random intercept.
[b] Models have been adjusted for a hyena's sex and level of human disturbance during their birth year as well as age (months), reproductive state among females, and the time of day when the fecal sample was collected.

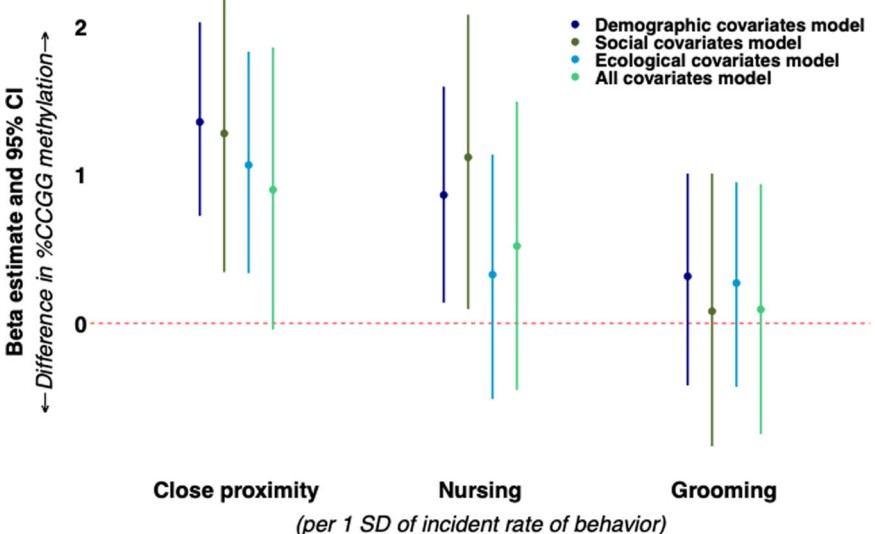

**Fig. 1 Associations of maternal care behaviors with cub and subadult offspring %CCGG methylation among $N = 55$ hyenas.** All models included a random intercept for maternal ID to account for relatedness and the offspring's age in months at the time of blood sample collection for DNA methylation quantification. Models were further adjusted according to four types of covariates (demographic covariates [dark blue], early social experience covariates [dark green], ecological covariates [light blue], and a fully adjusted all covariates model [light green]). 95% CIs were based on percentile bootstrapping (2000 iterations). Source data are provided as a Source Data file.

Next, we assessed relationships between social network metrics derived during the hyenas' CD and DI periods of development and %CCGG methylation. During the CD period, strength (0.75 [95% CI: 0.01, 1.48]) and betweenness (0.83 [95% CI: 0.10, 1.58]) were each positively associated with %CCGG methylation in demographic covariates models (Fig. 2). These relationships were attenuated after adjusting for early social experience and ecological covariates. Network degree was not associated with %CCGG methylation in any of our models during the CD period and none of the interaction terms involving social network metrics and maternal social rank were significant.

During the DI period, network degree was positively associated with %CCGG. Each 1-SD increment in degree corresponded with 0.69 (95% CI: 0.03, 1.38) higher %CCGG methylation in the demographic covariates model. This association was strengthened

after accounting for ecological covariates (1.04 [95% CI: 0.25, 1.86] %CCGG methylation), and also in the fully adjusted model that included all covariates (1.19 [95% CI: 0.24, 2.13] %CCGG methylation). Estimates for network degree from the social experience model included the null (0.79 [95% CI: −0.09, 1.69] %CCGG methylation, Fig. 3), although the direction and magnitude of estimates were similar to that of other models. Greater network strength was associated with 0.71 (95% CI: 0.03, 1.43) higher %CCGG methylation in the demographic covariates model. This estimate was materially unchanged after accounting for ecological covariates (0.68 [95% CI: −0.05, 1.40] %CCGG methylation) and adjustment for all covariates simultaneously (0.83 [95% CI: −0.03, 1.72] %CCGG methylation), although we note that the 95% CIs were slightly wider in these models and narrowly crossed the null (Fig. 3). Adjustment for social

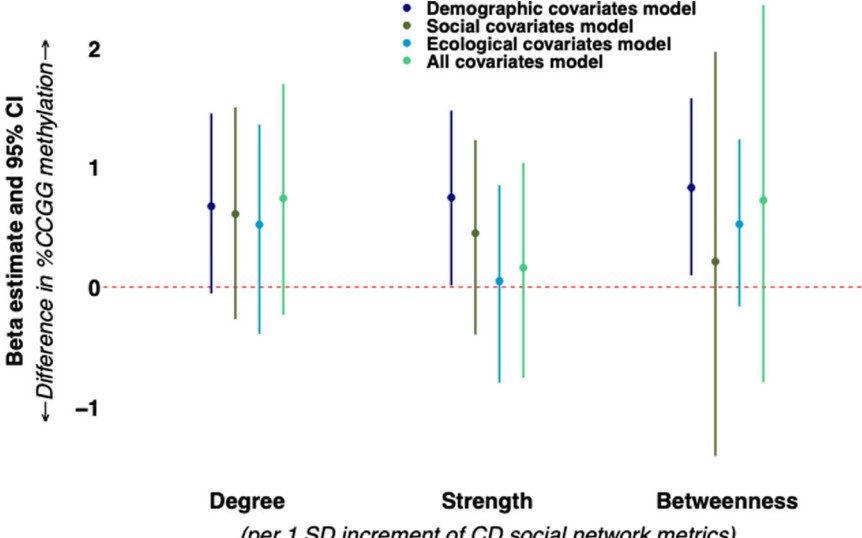

**Fig. 2 Associations of hyena CD social network metrics with cub and subadult offspring %CCGG methylation among *N* = 81 hyenas.** All models included a random intercept for maternal ID to account for relatedness, and the offspring's age in months at the time of blood sample collection for DNA methylation quantification. Models were further adjusted according to four types of covariates (demographic covariates [dark blue], early social experience covariates [dark green], ecological covariates [light blue], and a fully adjusted all covariates model [light green]). 95% CIs were based on percentile bootstrapping (2000 iterations). Source data are provided as a Source Data file.

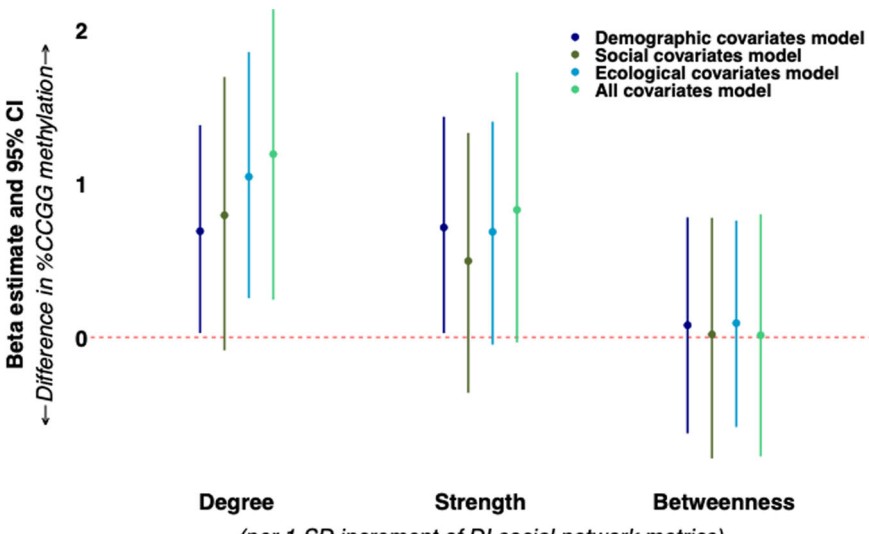

**Fig. 3 Associations of hyena DI social network metrics with cub and subadult offspring %CCGG methylation among *N* = 81 hyenas.** All models included a random intercept for maternal ID to account for relatedness, and the offspring's age in months at the time of blood sample collection for DNA methylation quantification. Models were further adjusted according to four types of covariates (demographic covariates [dark blue], early social experience covariates [dark green], ecological covariates [light blue], and a fully adjusted all covariates model [light green]). 95% CIs were based on percentile bootstrapping (2000 iterations). Source data are provided as a Source Data file.

experience covariates attenuated the estimate for network strength towards the null (0.49 [95% CI: −0.36, 1.33] %CCGG methylation). Betweenness was not associated with %CCGG methylation, and there were no significant interactions between DI network metrics and maternal rank.

Finally, offspring %CCGG methylation was not associated with adult fGCMs.

**Part 3: formal mediation analysis.** In the subset of hyenas for which we had data on maternal care (*n* = 30 total; *n* = 17 females and *n* = 13 males) or social network metrics (*n* = 52 total; *n* = 32 females and *n* = 20 males), as well as %CCGG DNA methylation and adult fGCMs, we conducted a three-step mediation analyses.

We found no association between our explanatory variables (maternal care, CD social connectedness, DI social connectedness) and the outcome, fGCMs (Supplementary Table 7). In the second step of the mediation analyses, we found no evidence that social experiences (the explanatory variables) were associated with %CCGG methylation (the mediator), or that %CCGG methylation was associated with fGCMs (the outcome variable, Supplementary Table 7). Given the lack of associations in the first two steps, we did not perform the final step of the mediation analysis.

**Part 4: genome-wide (multiplexed enhanced reduced representation bisulfite sequencing (mERRBS)) DNA methylation analyses.** Our epigenome-wide association study (EWAS)

resulted in 18 differentially methylated sites (DMSs) that were associated with fGCMs after false discovery rate (FDR) correction (Table 2 and Supplementary Fig. 10). While we did not observe systemic bias in the enrichment of positive or negative associations from our EWAS, within particular genomic regions, including CpG islands, promoters and introns there appeared to be fewer *Beta* estimates near zero than compared to all CpG sites across the whole genome (Supplementary Fig. 11). Based on the CpG density annotation, we observed that 62% of CpG sites from our EWAS occurred in CpG islands and CpG shores, while among both positive and negative DMSs, 72% of sites were in CpG islands and shores (Supplementary Fig. 12a, c). According to our genic annotation, 38% of CpG sites from our EWAS occurred in either promoters, exons, or introns, and 39% of DMSs were found in the same genic annotations (Supplementary Fig. 12b, d). There were equal total numbers of positive and negative DMSs, but in both annotations more positive DMSs were found near CpG islands and gene bodies than in intergenic regions.

Among the 18 CpG sites identified in the EWAS, we examined the relationships between maternal care metrics and maternal rank with DNA methylation and identified six CpG sites that were associated with maternal care. Four of these CpG sites exhibited positive *Beta* estimates from fGCMs models and corresponding negative *Beta* estimates for maternal grooming, while DNA methylation at one CpG site was negatively associated with fGCMs and positively associated with maternal grooming. At a CpG site located at the human teashirt zinc finger homeobox 2 (*TSHZ2*) gene[23,24], the proportion of time that cubs were groomed by their mothers was positively associated with DNA methylation ($P = 0.074$; Table 3), and fGCMs were negatively associated with DNA methylation ($P = 0.031$; Table 2). We observed a negative association of time spent grooming ($P < 0.001$; Table 3) and a positive association of fGCMs ($P = 0.026$; Table 2) with respect to DNA methylation at a CpG that aligned at the tsukushi, small leucine-rich proteoglycan (*TSKU*) gene in humans[23,24]. We also observed a negative association for time spent grooming ($P = 0.097$; Table 3) and a positive association of fGCMs ($P = 0.045$; Table 2) with DNA methylation at a CpG that aligned at the RAB43, member RAS oncogene family (*RAB43*) and the ISY1 splicing factor homolog (*ISY1*) gene complex in humans[23,24]. Maternal grooming was negatively associated with DNA methylation ($P = 0.048$; Table 3) and fGCMs were positively associated with DNA methylation ($P = 0.007$; Table 2) at a CpG site at the human adenosine deaminase (*ADA*) gene[23,24]. Similarly, we observed a negative association for time spent grooming ($P = 0.018$; Table 3) and a positive association for fGCMs ($P = 0.026$; Table 2) with DNA methylation at a CpG site at zinc finger and SCAN domain containing 2 (*ZSCAN2*) gene in humans[23,24].

In addition, one CpG site near the human DENN domain containing 3 (*DENND3*) gene[23,24] was associated with the proportion of time spent in close proximity to the mother hyena ($P = 0.042$; Table 3) and fGCMs ($P = 0.032$; Table 2), but the *Beta* estimates were in the same direction, and thus did not align with our criteria for a plausible mechanism or biomarker of a mechanism linking early social environment to future stress phenotype. Neither maternal rank during the year in which the offspring was born nor the proportion of time spent nursing was associated with the stress-related DMSs identified in this sample of female hyenas (Table 3).

**Candidate gene DNA methylation: putative GR promoter.** Among 78 hyenas that passed the pyrosequencing quality checks (Pyromark Q96 MD instrument), preliminary data revealed invariant and near-zero percent methylation at six CpG sites,

including the putative *NGFI-A* transcription factor binding site (Supplementary Fig. 13). Due to lack of variation in DNA methylation at these CpG sites, we did not conduct any additional analyses with respect to this region

## Discussion

Leveraging ecological and behavioral data from a long-term prospective study of wild hyenas, we investigated the extent to which social experiences during two early developmental windows are associated with DNA methylation and adult stress phenotype. We found beneficial effects of social connectedness during the DI period of development, when young hyenas no longer shelter in dens, on adult stress phenotype. We also found that higher-quality maternal care in the first year of life, which roughly corresponds with the CD period of development, and greater social connectedness during the DI period were each associated with higher global DNA methylation. Although we found no evidence that the relationship between social experience and stress phenotype is mediated by global DNA methylation, our genome-wide DNA methylation analysis identified multiple DMSs associated with both receipt of maternal care and adult fGCMs, and thus might serve as mechanisms or biomarkers of mechanisms for future investigation.

### The early social environment and adult stress phenotype

*Higher social connectedness is associated with lower fGCMs.* Greater social connectedness—specifically, higher degree centrality and strength—during the DI period, when hyenas have left the CD, corresponded with lower adult fGCMs. This supports the hypothesis that social connections promote a healthy stress response and suggests that connections with group members are particularly important among subadult animals as they expand their social network beyond a limited number of associations with group members at the CD.

Our findings corroborate previous work in humans, rodents, and baboons. For example, in young adults, Kornienko et al. showed that more gregarious individuals have lower salivary cortisol levels[25,26]. Similarly, Ponzi et al. reported that a children's perception of their social connections (i.e., higher density of friendships) is inversely related to salivary cortisol levels, as well as salivary alpha-amylase reactivity, an indicator of reactivity to psychological stressors[27]. In Sprague–Dawley rats, postweaning rearing in social isolation corresponded with higher corticosterone concentrations than in rats reared with other group members[6]. The importance of social networks has also been observed in wild baboons, for which membership in more close-knit and focused grooming networks are associated with lower fGCMs[28]. However, it is worth noting that social connections are not universally associated with stress phenotypes. For instance, a study on wild yellow-bellied marmots found no association of social network metrics with fGCMs measured in a mixed-age, mixed-sex population[29], although it should be noted that this species is not obligately social like those in other above-mentioned studies.

*Maternal care is not associated with fGCMs.* Contrary to our predictions and to findings from rodents, non-human primates, and humans[3,4,7,30,31], we found no evidence that maternal care affects future stress phenotype. There are multiple possible explanations for our null findings. First, in contrast to controlled rodent experiments, wild hyenas are subject to a multitude of stressors over development, which may hamper our ability to isolate the specific effect of maternal care on the stress phenotype. Second, compared to maternal separation contrived in many experimental studies, natural variation in maternal care is far

**Table 2 Epigenome-wide association results showing DMSs associated with fGCMs BLUPs among N = 25 hyenas.**

| CpG location in C. crocuta genome (scaffold #) | β ± SE† | FDR P value‡ | P† | Nearest human gene(s)a | Approx. location c | CHRc | Human ref. BLAT score/ % identical match/# bp span a | Cat ref. BLAT score/ % identical match/# bp span b | Gene name a |
|---|---|---|---|---|---|---|---|---|---|
| 119.9138483 | 0.118 ± 0.024 | 0.031721 | 7.167E−07 | DENND3 | Gene body | 8 | 700/87.8%/20,675 | 6691/87.9%/26,590 | DENN domain containing 3 |
| 27.4578142 | −0.022 ± 0.004 | 0.031432 | 6.628E−07 | TSHZ2 | Gene body | 20 | 4664/87.8%/23,955 | 12,602/88.8%/21,036 | Teashirt zinc finger homeobox 2 |
| 271.106661 | 0.023 ± 0.004 | 0.026112 | 4.518E−07 | TSKU | Gene body | 11 | 3511/85.5%/18,366 | 12,756/88.8%/19,971 | Tsukushi, small leucine-rich proteoglycan |
| 296.402426 | 0.026 ± 0.005 | 0.044999 | 1.22E−06 | RAB43/ISY1 | Gene body | 3 | 1109/87.3%/17,857 | 8296/88.0%/24,384 | RAB43, member RAS oncogene family / ISY1 splicing factor homolog |
| 477.763333 | 0.038 ± 0.007 | 0.006652 | 5.01E−08 | ADA | Gene body | 20 | 1996/87.6%/14,038 | 11,524/89.3%/17,915 | Adenosine deaminase |
| 7.745562 | 0.018 ± 0.004 | 0.026112 | 5.113E−07 | ZSCAN2 | Gene body | 15 | 1311/85.6%/11,881 | 11,637/89.3%/19,302 | Zinc finger and SCAN domain containing 2 |

Only DMSs that were also significantly associated with a maternal care metric at P < 0.1 are shown in this table, while the remaining fGCM-associated DMSs are in Supplementary Table 8. DNA methylation was assessed during the subadult life stage, and fecal samples were collected from animals during the subadult and adult life stages.
DMSs differentially methylated CpG sites, fGCMs fecal glucocorticoid metabolites, BLUPs best linear unbiased predictors, CHR chromosome.
† Beta estimates and unadjusted P values are from MACAU and then corrected for inflation and bias using the package bacon.
‡ FDR P value based on Benjamini–Hochberg correction (5%).
a Mapping of spotted hyena DNA sequences spanning 20.4k bp onto the human genome (Human Dec. 2013 [GRCh38/hg38] Assembly) was done using the top hit from the UCSC Genome Browser's BLAT tool.
b Mapping of spotted hyena DNA sequences spanning 20.4k bp onto the domestic cat genome (Felis catus Nov. 2017 [felCat9] Assembly) was done using the top hit from the UCSC Genome Browser's BLAT tool.
c Reference genome is hg38.

**Table 3 Associations of DMSs by maternal rank and maternal care variables among the 18 DMSs identified in the stress phenotype EWAS for female spotted hyenas that have (maternal rank data ($n$ = 23) and maternal care data ($n$ = 9).**

| Closest human gene [s] (BLAT)[a] | CpG site location in C. crocuta genome (scaffold #) | Maternal rank $\beta \pm SE^b$ | $p^b$ | Close proximity $\beta \pm SE^b$ | $P^b$ | Nursing $\beta \pm SE^b$ | $p^b$ | Grooming $\beta \pm SEc^b$ | $p^b$ | $\beta$ sign (±) matches prediction?[c] |
|---|---|---|---|---|---|---|---|---|---|---|
| DENND3 | 119.9138483 | −0.234 ± 0.153[b] | 0.125 | 6.407 ± 3.156[b] | 0.042 | −4.930 ± 3.948[b] | 0.212 | −1.524 ± 2.224[b] | 0.493 | No |
| MROH5/TSNARE1 | 119.9561382 | 0.033 ± 0.045 | 0.463 | −0.788 ± 0.639 | 0.218 | 0.220 ± 0.728 | 0.763 | 0.267 ± 0.446 | 0.550 | |
| NA | 124.5690667 | −0.018 ± 0.078[b] | 0.813 | −0.654 ± 1.272[b] | 0.607 | 0.174 ± 1.368[b] | 0.899 | 0.747 ± 0.807[b] | 0.354 | |
| NA | 221.28127 | −0.012 ± 0.099[b] | 0.903 | 1.107 ± 1.020 | 0.278 | −0.277 ± 1.006 | 0.783 | −0.366 ± 0.630 | 0.562 | |
| MAPK8IP3/JPT2 | 234.3124311 | −0.050 ± 0.076 | 0.513 | 0.405 ± 1.022 | 0.691 | −0.831 ± 1.128 | 0.461 | −0.266 ± 0.650 | 0.683 | |
| TSHZ2 | 27.4578142 | −0.057 ± 0.070 | 0.415 | −0.200 ± 0.994 | 0.841 | 1.345 ± 1.148 | 0.241 | 1.139 ± 0.638 | 0.074 | Yes |
| TSKU | 271.106661 | 0.029 ± 0.120[b] | 0.808 | −0.046 ± 1.697[b] | 0.979 | −2.206 ± 1.693[b] | 0.192 | −2.506 ± 0.680 | <0.001 | Yes |
| RADIL | 273.4081006 | 0.308 ± 0.777[b] | 0.692 | −13.043 ± 22.046[b] | 0.554 | −1.765 ± 21.243[b] | 0.934 | −10.746 ± 13.317[b] | 0.420 | |
| ZMIZ1 | 28.1531871 | −0.082 ± 0.071 | 0.246 | 0.957 ± 1.225 | 0.435 | −1.149 ± 1.084 | 0.289 | −0.863 ± 0.637 | 0.176 | Yes |
| RAB43/ISY1 | 296.402426 | 0.028 ± 0.057 | 0.622 | −0.630 ± 0.746 | 0.398 | −1.286 ± 0.957 | 0.179 | −0.936 ± 0.564 | 0.097 | |
| PITRM1 | 32.11390064 | −0.028 ± 0.084 | 0.740 | 0.160 ± 1.200 | 0.894 | −2.342 ± 1.514 | 0.122 | −0.846 ± 0.922 | 0.359 | |
| MYBPC2 | 334.155724 | −0.071 ± 0.095[b] | 0.457 | −1.301 ± 1.350[b] | 0.335 | 1.363 ± 1.528[b] | 0.372 | 1.267 ± 0.862[b] | 0.142 | |
| CRNN | 415.2139465 | −0.581 ± 0.780[b] | 0.456 | −6.303 ± 22.917[b] | 0.783 | −2.224 ± 15.088[b] | 0.883 | −17.775 ± 28.750[b] | 0.536 | |
| PLPPR3/AZU1/PRTN3/ELANE | 443.367667 | −0.097 ± 0.260[b] | 0.710 | −1.287 ± 1.609 | 0.424 | 0.204 ± 1.796 | 0.910 | 1.377 ± 1.086 | 0.205 | |
| ADA | 477.763333 | −0.133 ± 0.291[b] | 0.648 | 4.234 ± 3.865[b] | 0.273 | −2.415 ± 4.561[b] | 0.596 | −4.302 ± 2.172[b] | 0.048 | Yes |
| MACROD1/FLRT1 | 53.1319257 | −0.026 ± 0.139[b] | 0.851 | −0.405 ± 3.863[b] | 0.916 | 0.478 ± 3.772[b] | 0.899 | 0.979 ± 2.325[b] | 0.674 | |
| SOWAHA/SHROOM1 | 64.5896792 | −0.014 ± 0.129[b] | 0.915 | −2.345 ± 2.410[b] | 0.330 | 0.329 ± 2.604[b] | 0.900 | 0.579 ± 1.572[b] | 0.713 | |
| ZSCAN2 | 7.745562 | 0.065 ± 0.108[b] | 0.550 | 0.546 ± 2.076[b] | 0.792 | −3.079 ± 1.979[b] | 0.120 | −2.445 ± 1.030[b] | 0.018 | Yes |

DMSs differentially methylated CpG sites, EWAS epigenome-wide association study.
[a] Mapping of spotted hyena DNA sequences onto the human genome (Human Dec. 2013 [GRCh38/hg38] Assembly) via the UCSC Genome Browser's BLAT tool.
[b] If overdispersion was detected, then a negative binomial model was fit instead of a Poisson model.
[c] The Beta estimates from the fecal corticosterone EWAS and from generalized linear regression models in which maternal rank and maternal care variables were the independent variables, were expected to be in opposite directions; "yes" demarks where this expectation was met.
Untransformed bBeta estimates are from a Poisson model in which methylated read counts are the outcome and total read counts are included as an offset. Models control for hyena age in months at the time of the darting.

more subtle and may require a more sensitive measure of physiological stress than average fGCMs, which is a summary baseline stress indictor[32] subject to unmeasured environmental factors and the animal's condition when the sample is collected[33]. Considering not only the discrepancies in how early-life experience and stress outcomes are measured but also differences between controlled laboratory settings vs. studies of wild animals, we urge caution when comparing our results to previous studies involving captive primates and rodents.

### The early social environment and global DNA methylation
*Maternal care and social connectivity are associated with global DNA methylation.* Greater maternal care (specifically, close maternal proximity during the first year of life) and higher social connectivity (namely, degree during the DI life stage) were both associated with higher global DNA methylation later in life. These results corroborate those of previous studies that report associations of early social experiences with differential patterns of DNA methylation in rodents, rhesus macaques, and humans[11–13]. Specifically, offspring that spent more time in close proximity to their mothers during the first year of life had higher global DNA methylation later in life, even after adjustment for key demographic, social, and ecological covariates. We also found that more time spent nursing corresponded to higher global DNA methylation after adjusting for demographic and social experience covariates, although the effect of nursing was attenuated when we included the early-life ecological covariates in the model. This suggests that ecological context may account for some variation in the relationship between time spent nursing and global DNA methylation, which is plausible given previously reported associations in hyenas between maternal care behavior as well global DNA methylation with human disturbance and prey availability[34,35]. One potential explanation for the robust effect of close proximity, but not the other metrics of maternal care (nursing and grooming), on global DNA methylation may reflect the fact that close proximity includes the time mother hyenas spent nursing and grooming. Thus, close proximity is a composite metric of maternal investment in offspring that may capture more variation in offspring global DNA methylation.

We also found that higher social connectedness during the DI, but not CD, period of development was positively associated with global DNA methylation. Specifically, the degree of a hyena's social connectedness, which reflects the number of group members with which a hyena associates, corresponded with higher global DNA methylation even after adjustment for demographic and ecological covariates. Controlling for early-life social factors, including the number of littermates, group size, and maternal rank slightly attenuated the effect of social connections towards the null, which is not surprising given that these variables are known determinants of a young hyena's social network[36] as well as of global DNA methylation[34].

Our findings that less maternal care during the first year of life and less social connectedness during the DI period of development are associated with lower global DNA methylation suggest that inadequate early social experiences may be risk factors for suboptimal health later in life. Previous human research has shown that lower global DNA methylation corresponds with adverse health outcomes related to cardiovascular disease[37], pulmonary disease[38], shorter telomere length (a marker of accelerated aging) in both adolescents[39] and adults[40], and cancer[41]. It is worth noting that higher global DNA methylation has also been associated with some cardiovascular diseases[42]. Thus, deviations in either direction, both lower and higher DNA methylation, may be a marker of aberrant physiology and should be interpreted with caution. Finally, the effect sizes for %CCGG

methylation we observed are modest but biologically relevant, as they are comparable in magnitude to the effect sizes of dietary[43] and toxicant[44] exposures reported in the existing literature.

### Mediation by global DNA methylation
*No evidence of mediation by global DNA methylation.* We found no evidence of mediation by %CCGG methylation. This could be due to low power given the relatively small sample sizes, and/or related to the fact that %CCGG methylation averaged over the entire genome may be too broad a metric to isolate important regulatory pathways involved in the stress response. For example, in humans, we previously found that maternal prenatal socioeconomic status was associated with differential methylation at specific regions of the genome, but not with global DNA methylation in humans[45]. Therefore, we complemented our global DNA methylation analysis in hyenas with mERRBS[46], a high-throughput sequencing technology that assays single-nucleotide DNA methylation resolution at a genome-wide scale.

### Biological interpretation of CpG sites identified in genome-wide DNA methylation assessment. Using mERRBS, we identified 18 DMSs that were associated with fGCMs after FDR correction, six of which were also associated with early-life maternal care metrics. Five of these DMSs followed our prediction given that greater maternal care is hypothesized to be protective against an adverse stress phenotype. More specifically, for a given DMS, a positive association between DNA methylation and stress hormone levels should correspond to a negative association between maternal care and DNA methylation at that DMS, or vice versa. We discuss the direction of the associations of these five DMSs with respect to fGCMs and early-life maternal care metrics below, as well as their potential biological function.

DNA methylation of a CpG site that mapped near the human *TSHZ2* gene was positively associated with receipt of maternal grooming and inversely associated with fGCMs. This transcription factor is involved in regulating gene expression and plays a role in development in multicellular organisms[47–50]. *TSHZ2* is expressed during early development of the nervous system in diverse vertebrate taxa[51,52]. The *TSHZ2* gene is also associated with a number of pathophysiological states, including inflammatory disease[53], dysregulated sleep patterns[54], and tumorigenesis[55,56] via hypomethylation of *TSHZ2* promoter in cancerous epithelia cell lines[57]. Second, we identified a CpG site that mapped near the human *TSKU* gene which was negatively associated with receipt of maternal grooming and positively associated with fGCMs. *TSKU* is part of gene ontogeny pathways involved in gene expression and development as well as differentiation of the ectoderm germ layer, wound healing, *Wnt* signaling, and cholesterol processing[47–50]. Expression of *TSKU* has been implicated in renal disease through fibrosis and inflammation[58]. The third CpG site, which was negatively associated with grooming and positively associated with fGCMs, mapped near the *RAB43* and *ISY1* genes. *RAB43* plays a role in phagosome maturation, response to interferon-gamma, Golgi organization, and intracellular protein transport, while *ISY1* contributes to messenger RNA (mRNA) processing and splicing as well as transcription-coupled nucleotide-excision repair[47–50]. As a regulator of cellular trafficking and signaling, there is evidence that *RAB43* helps orchestrate the immune response to infection[59,60]. More recent work also implicates overexpression of this gene in the progression of multiple cancers[61,62]. The fourth CpG site exhibited a negative association with grooming and a positive association with fGCMs, and it mapped to the *ADA* gene, which is involved in T and B cell regulation, inflammation, aging, embryonic development, and organic molecule metabolism[47–50]. Research in humans and rodent models of human health

suggests that insufficient expression of *ADA* gene leads to immunodeficiency[63–65] and *ADA* activity has been identified as an indicator of aging[66–69]. Of particular interest to our study, corticosterone has been shown to inhibit *ADA* activity in mice[70], suggesting that regulation of this gene may link early-life experience with stress phenotype and aging. Finally, DNA methylation of a CpG site near the *ZSCAN2* gene was negatively associated with receipt of grooming and positively associated with fGCMs. *ZSCAN2* is involved in the regulation of transcription and cell differentiation, particularly during development[47–50], but we found no reported pathologies associated with this gene. Despite having only correlative evidence from our EWAS, the above molecular biomarkers in hyenas share sequence similarities to human genes. Furthermore, these genes have been implicated in basic biological processes related to a range of chronic diseases including inflammation, immune function, tumorigenesis, and aging, suggesting that these pathways may be particularly important for understanding the relationship between early-life maternal care and later-life stress phenotypes.

**No evidence of DNA methylation variation at the putative hyena GR**. Although some previous work in rodents and humans has shown that inadequate maternal care and greater early adversity corresponded with higher methylation of CpG sites in the GR promoter region[10,71], this relationship between maternal care and GR DNA methylation has not been replicated in all studies[72]. In the present study, we measured DNA methylation at CpG sites in the putative hyena *GR* gene promoter and found invariant and near-zero percent methylation, suggesting a potential lack of plasticity in DNA methylation in this promoter region, at least when using peripheral leukocytes. Future studies may better replicate results from laboratory rodents by using brain tissue, wherein expression of the *GR* gene is more tightly coupled with HPA regulation, or by exploring DNA methylation of additional CpGs with more extensive coverage of the hyena *GR* gene.

**Limitations of this study**. Our study has several limitations. First, we used a large database of existing behavioral, demographic, and biological variables to test our hypotheses. As a result, we lacked complete data overlap between different variables of interest resulting in somewhat differing sample sizes across analyses (thereby hindering our ability to make direct comparisons across analytical subsamples). Further, the observational nature of the data limits causal inference due to the potential for unmeasured confounding. Next, our genome-wide analysis was limited by budgetary constraints, so sample selection focused on animals of similar age and sex for the sake of statistical power. Consequently, results of this particular analysis may not be generalizable to both sexes and preclude between sex comparisons. We acknowledge the tradeoff between improved statistical power and internal validity at the expense of external validity of results given the opportunistic nature of this analysis. Second, we used the average of multiple measures of fGCMs as an indicator of the animals' stress phenotype. However, a more nuanced measure of stress, like reaction and recovery to a standardized stressor or the ratio of cortisol to dehydroepiandrosterone, may provide a more direct assessment of acute HPA function[73,74] over time. Third, our mediation and mERRBS analyses comprised small sample sizes and therefore may have low external validity—particularly for the mERRBS analysis, which included only females. Fourth, we used archived DNA from whole blood that lacked information on cell type composition, and therefore we were not able to control for cell-type heterogeneity as a source of variation in DNA methylation. However, cell-type composition may be influenced by factors like social stress[75], and therefore might be on the causal pathway between the explanatory variables of interest and DNA methylation. Future analyses that aim to test hypotheses about functional genomic pathways involving a larger sample of genome-wide DNA methylation data are warranted. With larger sample sizes and additional annotation of the hyena genome, future studies could home in on specific genomic contexts containing differentially methylated CpG sites and identify pathways enriched in response to early-life social experience. Finally, we recommend future studies incorporate low coverage genome-wide genetic sequence data as a more accurate alternative to a pedigree to control for genetic correlations in DNA methylation.

In conclusion, in a wild population of spotted hyenas, maternal care and social connections early in life were key determinants of global DNA methylation and fGCMs assessed in adulthood, even after accounting for key demographic, biological, and social factors. These findings contribute to the literature in multiple ways. First, few studies have been able to assess early social experience, DNA methylation, and future stress phenotype together, a study design that allows for explicit assessment of DNA methylation as a potential mediator. Second, existing literature has focused predominantly on singular social experiences during specific developmental windows, which is not only unrealistic given the multitude of experiences across early life that may contribute to stress phenotypes but also precludes the ability to assess the relative importance of the type and timing of experiences. In the present study, we were able to examine prospective associations of multiple aspects of social experience during two sensitive windows of development.

Our work on wild hyenas joins a growing body of evidence, predominantly from rodents and primates, showing the critical effects of early experience on molecular biomarkers and future stress phenotype. Given that such studies are seldom possible in free-living populations of long-lived social animals, this study provides insight into the type, timing, and mechanisms that link early-life experiences with the developmental plasticity of stress phenotypes and highlights the extent to which natural variation in maternal care and social interactions have persistent effects on developing offspring.

## Methods

**Study population**. We used behavioral data and biological samples collected between June 1988 and July 2016 by the Mara Hyena Project, an ongoing field study of wild spotted hyenas (*C. crocuta*) in the Masai Mara National Reserve, Kenya. All protocols used for the present study have been approved by Michigan State University (MSU) Institutional Animal Care and Use Committee (IACUC) and the Kenyan Wildlife Service (KWS). For each hyena in our study population, we have information on demographic, social, and ecological conditions throughout their lifetimes. Using blood samples from opportunistically immobilized hyenas paired with archived behavioral and physiological data, we constructed three primary datasets for analyses: global DNA methylation ($n = 186$ total; $n = 99$ females and $n = 87$ males), genome-wide DNA methylation ($n = 29$ total; $n = 29$ females and $n = 0$ males), and candidate gene DNA methylation ($n = 78$ total; $n = 43$ females and $n = 35$ males).

Spotted hyenas are an appropriate species in which to test our hypotheses because they exhibit a wide range of social behaviors, live in large fission–fusion clans that can contain more than 100 individuals, and show a protracted period of maternal care[76]. Female hyenas reach reproductive maturity around 2 years of age and give birth to one or two offspring every 14–17 months starting in their third year of life[77]. Following a 110-day gestation period, hyena cubs spend the first few weeks of life interacting exclusively with their mothers at a natal den[78]. During development, young hyenas rely on their mothers for sustenance and social support until they are at least 2 years of age[78]. Young hyenas also socialize with other members of their clan during an earlier life stage when they reside at their clan's CD—known as the CD period, followed by a later, DI period when hyenas venture out into their clan's territory and develop relationships with other group members[36,79]. In order to balance our sampling design, we matched the duration of the DI period with that of the CD period (mean length of both periods was $6.98 \pm 1.74$ months)[80] for each hyena.

**Early social experiences**. We derived the maternal care variables from focal animal survey (FAS) data collected during observation sessions in which: (1) mother–offspring pairs were present together for a minimum of 5 min and offspring were <13 months old and (2) mothers were lactating since our intention was to focus on maternal care received early in life while offspring were dependent on nursing for sustenance. We quantified durations of maternal care behaviors from FAS data[81] based on counts of behaviors occurring during each minute of observation in which both the mother and offspring were present together. Behavioral data were collected daily between roughly 0530–0900 and 1700–2000 hours. We focused on three maternal care behaviors: minutes the mother and cub spent in close proximity (≤1 m apart), minutes offspring spent latched to their mother's nipple (nursing), and minutes during which mothers were observed grooming (i.e., licking) their offspring. Information on additional behavioral data processing appears in the Supplementary Material.

We measured social connectivity by generating association networks among hyenas based on co-occurrences between each hyena and its group members[36] as described in the Supplementary Material. For each hyena, we constructed separate association networks during the CD and DI periods (two life stages during which social interactions were previously identified as key determinants of fitness[80]).

From each hyena's association networks during its CD and DI periods, we extracted three metrics that quantify how connected a hyena is with its group members: degree centrality, strength, and betweenness centrality. We focused on these metrics, rather than the overall network structure, because they reflect an individual hyena's connectedness within its network. A description of social network methods has been published in ref. [36] and is summarized in the Supplementary Material.

**DNA methylation: global (%CCGG), genome-wide, and candidate gene**. We quantified global (%CCGG) DNA methylation derived from whole blood using the LUminometric Methylation Assay (LUMA)[82]. Descriptions of the LUMA assay, laboratory procedures, and data cleaning protocol are available in ref. [34], from which the current global DNA methylation samples were drawn. The majority of CpG sites assessed via LUMA occur in intergenic regions of the genome, where they may repress repetitive elements[83] and enhance chromosome stability[84], as well as introns, where they may function in transcription regulation and alternative splicing[85] (Supplementary Fig. 8). Coupled with evidence that global DNA methylation is responsive to early-life environmental factors[86–88], and the fact that lower global DNA methylation is associated with a range of adverse health outcomes in general-risk human populations (e.g., shorter telomere length as a metric of accelerated aging[39,40]; hypertension[37]; and chronic obstructive pulmonary disease[38]), we interpret average %CCGG methylation as a relevant biomarker of developmental plasticity where lower global DNA methylation is disadvantageous to health.

We measured genome-wide DNA methylation at a single-nucleotide resolution in whole blood collected from hyenas 11–27 months old. To prepare the mERRBS sample library, we followed the protocol of Garrett-Bakelman et al.[89] using 100 ng of high-quality genomic DNA. Each sample was spiked with 1 ng of non-methylated lambda guide DNA to estimate bisulfite conversion efficiency. Five libraries were pooled per lane of an Illumina HiSeq4000® platform for single-end sequencing with a 50-nucleotide read length. After DNA sequencing, we ran a standard bioinformatic pipeline to clean, align, and call DNA methylation reads using the draft spotted hyena genome[90]. We filtered read counts to a minimum of 10× coverage. Details on DNA library preparation and the bioinformatics pipeline are given in the Supplementary Material.

Given the extensive literature on the relevance of the GR gene to both early social experiences and stress phenotypes, we also assessed CpG methylation in the putative GR promoter region of DNA from hyenas. Candidate gene bioinformatics and laboratory methods are described in the Supplementary Material, and a list of all primers used in the candidate gene analysis is provided in Supplementary Table 2. We identified CpG sites in hyena DNA that aligned with those in DNA from humans and rats (Supplementary Figs. 2–5; refs. [91,92]). We calculated CpG site-specific DNA methylation in the putative hyena GR promoter region using pyrosequencing.

**Fecal glucocorticoid metabolites**. Since January 1993, we have opportunistically collected fecal samples any time an individually identifiable hyena was seen defecating. Fecal samples were mixed and transferred to 2 mL cryovials before flash freezing in liquid nitrogen within 12 h of collection. The frozen samples were then transported from our field site to the United States. Here, we focus on the hormone corticosterone, as indicated by the concentration of fGCMs measured via a validated hormone extraction process and radioimmunoassay developed for our study population[93,94].

**Demographic, social experience, and ecological covariates**. We considered three categories of potential confounding variables (i.e., those that are associated with the explanatory variable but not caused by the explanatory variable and potential determinants of the dependent variable) and included them in multiple variable models to improve causal inference. Sex was a demographic confounder; maternal rank, litter size, parity, and clan size were social experience confounders;

and human disturbance and local prey abundance were ecological confounders. Prey abundance was defined as either high or low, respectively, based on the annual mass migration of wildebeest and zebra in the study area. The Supplementary Material provides details on data collection germane to these variables.

**Statistical modeling framework**. We analyzed data in four distinct parts in accordance with our goals and using a methodical approach to make causal inferences from observational data[95]. First, we characterized the relationship between early social experience according to maternal care and social network metrics (explanatory variables) and adult stress phenotype as indicated by adult fGCMs (dependent variable). Second, we examined associations between early social experience and global DNA methylation, and between global DNA methylation and fGCMs. Third, given that global DNA methylation may be a potential mechanism (in addition to an outcome), we conducted a formal mediation analysis following Baron and Kenny's method (1986) on a subset of hyenas (n = 30 maternal care and n = 52 social networks) for which we had data on early social experience, global DNA methylation, and adult fGCMs (Fig. 7). Fourth, to complement the third objective, we used genome-wide DNA methylation data to identify potential functional biomarkers that might link early-life experiences with future stress phenotypes and that might be formally assessed as mediators in future analyses. In all models, we considered α = 0.05 as the threshold for statistical significance, unless otherwise indicated.

**Part 1: associations among early social environment, DNA methylation, and adult fGCMs**. First, we examined each maternal care variable (proportion of time in close proximity, nursing, and grooming) and social network metric (degree centrality, strength, and betweenness centrality during the CD and DI periods) as explanatory variables in relation to fGCMs as the dependent variable using mixed-effects linear regression with a random intercept for hyena ID to account for correlations between repeated measures of fGCMs. In these models, we included covariates that are associated with the explanatory variable and are potential determinants of the dependent variable (confounders), as well as factors that account for variability in the exposure only or the outcome only (precision covariates). Confounders included offspring sex and level of human disturbance during the year in which offspring were born (low, medium, and high). Precision covariates included hyena's age, reproductive state for females (nulliparous, pregnant, lactating, or other), and time of day (a.m. or p.m.) when the fecal sample was collected (Fig. 4).

**Part 2: associations between early social environment and global DNA methylation, and between global DNA methylation and fGCMs**. First, we modeled the total effect each maternal behavior and social network metric had as separate explanatory variables with global DNA methylation (%CCGG methylation) as a continuous dependent variable, again using mixed-effects linear regression models with a random intercept for maternal ID to account for correlations among siblings, and included the covariates depicted in Fig. 5. In multiple variable analysis, we controlled for offspring age in months as a precision covariate in all models because age is a known determinant of DNA methylation in this and other species[34]. In models where a maternal care behavior was the explanatory variable, we further accounted for three categories of covariates using separate models: demographic covariates (offspring sex), social experience covariates (number of littermates, mother's parity, and mother's social rank), ecological covariates (human disturbance and prey abundance on the date offspring were born), and a fully adjusted model that included all covariates. When CD or DI social network metrics were the explanatory variables of interest, we controlled for the same confounders, except we included clan size rather than the mother's parity in the social experience covariates model. Given the importance of maternal rank to offspring DNA methylation[34], and the potential for maternal rank to mitigate or exacerbate effects of mother–offspring interactions[35,96], we tested for statistical interactions between the explanatory variables of interest (maternal care and social connectedness) and maternal rank in models where %CCGG methylation was the outcome. We considered α = 0.10 for the interaction term as evidence of effect modification by maternal rank.

Next, we assessed the relationship between global DNA methylation and adult fGCMs (Fig. 6). We used mixed-effects linear regression with a random effect for individual ID to account for repeated fGCM measurements and controlled for sex and human disturbance during the year offspring were born. Precision covariates included the hyena's age, reproductive state, and time of day when the fecal sample was collected.

Of note, one of our aims was to examine associations with DNA methylation of the GR promoter region among 78 hyenas with adequate high-quality DNA. However, our assays revealed invariant and near-zero percent methylation at six CpG sites, including the putative NGFI-A transcription factor binding site, so no additional analyses were performed with respect to this region.

**Part 3: formal mediation analysis**. The third part of our analysis assessed the extent to which maternal care and social network metrics during CD and subadult DI life stages are associated with adult fGCMs before and after adjustment for global DNA methylation (%CCGG) as a mediator. Here, we included the

subsample of hyenas with complete data on early social experience, DNA methylation, and stress phenotype. Using the complete data subsample, we followed Baron and Kenny's mediation procedure (1986) and re-implemented the Parts 1 and 2 from above, which correspond to Steps 1 and 2 in a mediation analysis (Fig. 7). If associations from Steps 1 and 2 were significant, then we conducted Step 3 where we included %CCGG as an additional covariate in models in which each early-life social variable was the explanatory variable and fGCMs was the outcome (Step 3; Fig. 7) and compared estimates of association for each early-life social variable with vs. without the inclusion of the DNA methylation mediator[97]. We considered evidence for mediation by DNA methylation if the inclusion of %CCGG methylation attenuated the estimate for a given early-life social metric by >10%.

**Part 4: genome-wide (mERRBS) DNA methylation analyses**. Finally, as a complement to Part 3, we sought to identify differentially methylated CpG sites

that were associated with later-life corticosterone levels (fGCMs) as well as maternal care and/or maternal social rank during a hyena's first year of life as a way to identify biomarkers of this relationship following the "meet-in-the-middle" approach for high-dimensional data[98]. For this exploratory analysis, we selected 30 female hyenas from a single social group, born between 2011 and 2014 (except for one hyena born in 2010 and another born in 2007), and ranging in age from 11 to 27 months old at the time of darting in order to reduce variability in background characteristics. Of the selected 30 hyenas, we had information on fGCMs during the subadult or adult life stages (≥13 months) for 25 hyenas; these individuals comprised the study sample for the EWAS (Supplementary Fig. 7).

Prior to formal analysis, we filtered the DNA methylation data by removing 5% of CpG sites with the lowest interindividual variation in DNA methylation and removing CpG sites for which the average DNA methylation was <10 or >90%[99]. Next, to prepare the fGCMs data for the EWAS, we consolidated the repeated measures (range 1–18) of fGCMs such that each individual had one value for their stress phenotype using a mixed-effects linear regression model where repeated measures of fGCMs were the

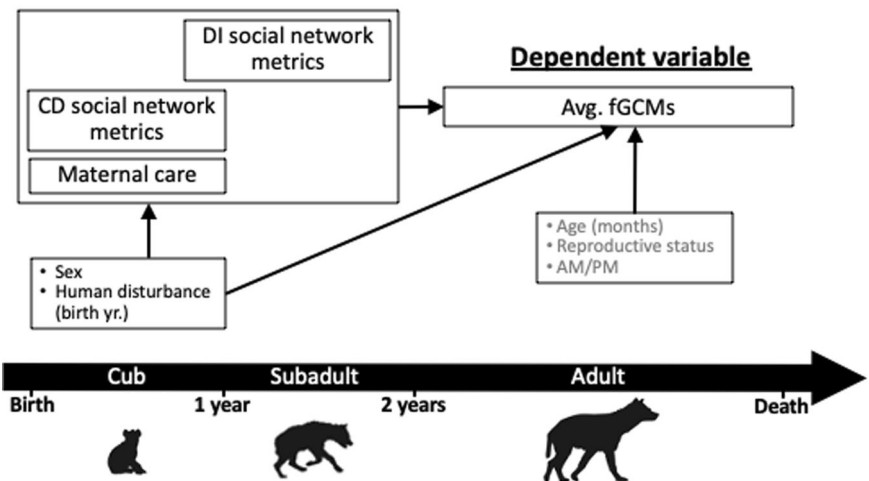

**Fig. 4 Directed acyclic graph (DAG) for Part 1 of the analysis showing the hypothesized relationships between early-life social experience and adult fecal glucocorticoid metabolites (fGCMs).** Covariates were included based on a priori biological knowledge and bivariate analysis (see Supplementary Tables 3–6). Precision covariates (gray text) were assessed when the fecal sample was collected. Variable positions over the timeline roughly corresponded with the timing of the assessment.

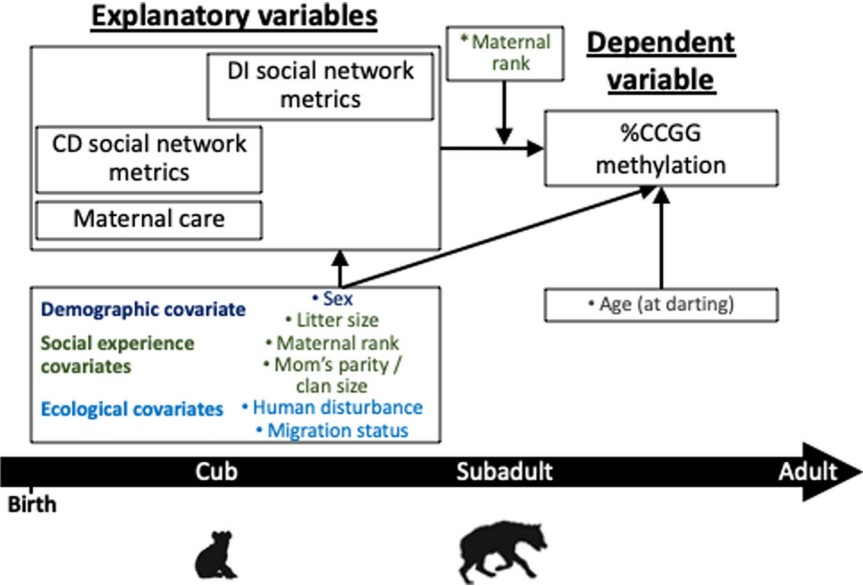

**Fig. 5 Directed acyclic graph (DAG) for Part 2 of the analysis showing the hypothesized relationships between early-life social experiences and % CCGG methylation.** Potential confounding variables were modeled in three different context groups, which included demographic covariates (dark blue), early social experience covariates (dark green), and ecological covariates (light blue). *Maternal rank was considered a potential effect modifier of the associations of interest. Age at darting was included as a precision covariate. Variable positions over the timeline roughly corresponded with the timing of the assessment.

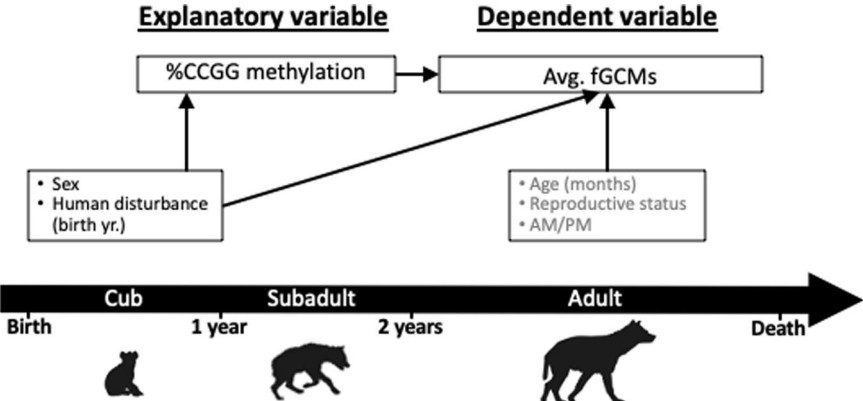

**Fig. 6 Directed acyclic graph (DAG) for Part 2 of the analysis showing the hypothesized relationships between %CCGG methylation with adult fecal Glucocorticoid Metabolites (fGCMs).** Covariates are included based on a priori biological knowledge and bivariate analysis (see Supplementary Tables 3–6). Precision covariates (gray text) are assessed when the fecal sample was collected. Variable positions over the timeline roughly correspond with the timing of the assessment.

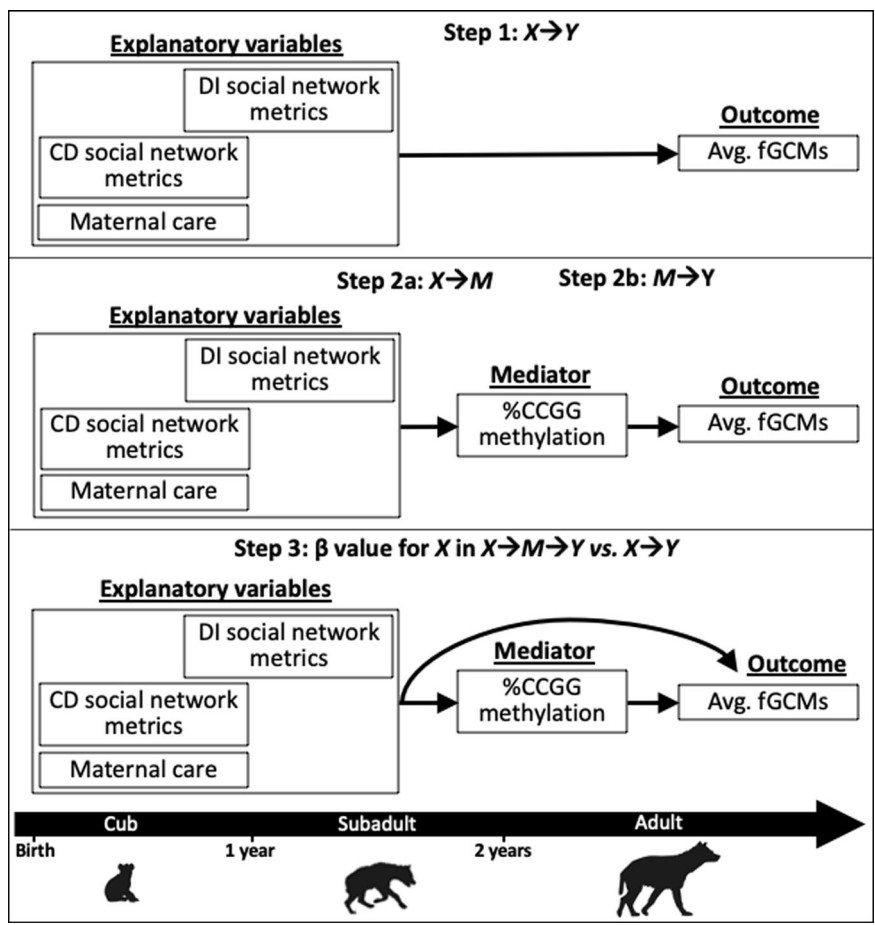

**Fig. 7 A conceptual diagram outlining the analytical strategy for the three-step mediation analysis.** *X* represents the explanatory variables of interest, *M* represents the potential mediator variable, and *Y* represents the outcome variable.

outcomes and a random intercept for hyena ID was included to output the best linear unbiased predictors (BLUPs) from this model (see Supplementary Material). These BLUPs represent the deviation of each individual's average fGCMs concentration (i.e., the mean across all available measurements) from the population average. We then analyzed associations of the BLUPs with genome-wide DNA methylation using binomial regression models in MACAU[100]. In these models, the BLUPs were the explanatory variable and DNA methylation of each sequenced CpG site (count of methylated divided by total CpG sites) was the outcome (Supplementary Fig. 7a). We controlled for offspring age in months at the time of darting as a precision covariate, and we accounted for genetic correlations by including a relatedness covariance matrix based

on a pedigree of our study population. After running the EWAS, we corrected estimates of association and *P* values for inflation and bias due to unmeasured confounding using the Bioconductor package, "bacon" in order to reduce false-positive associations[101]. We accounted for multiple comparisons using a Benjamini–Hochberg FDR of 5% on the inflation and bias-corrected *P* values[102]. Finally, among the 25 individuals for which we assayed mERRBS, we annotated CpG sites from the EWAS in genomic regions of the draft hyena genome as described in the Supplementary Material. We then assessed patterns of positive and negative associations of DNA methylation with fGCMs and compared the distribution of *Beta* estimates within genomic regions vs. across the entire genome to assess for bias in the direction of associations from the EWAS.

Among the DMSs identified in the EWAS, we focused on those that were also associated with maternal care and/or maternal rank. By focusing on CpG sites that are associated with later-life fGCMs and with maternal care and/or maternal rank, we identified potential biomarkers of stress-related genomic pathways responsive to early social experience (Supplementary Fig. 7b, c).

To accomplish this, we ran separate generalized linear regression models where the explanatory variable was a given maternal care metric or maternal rank, and the outcome was counts of methylated DNA sequence reads at each stress-related DMS identified from the EWAS. We treated the total number of DNA sequence read counts (methylated + unmethylated) at each DMS as an offset in our models. We considered a CpG site to be of interest based on two criteria. First, the *Beta* coefficient for the relationship between a given early social characteristic and DNA methylation for the CpG site of interest needed to reach statistical significance at a nominal *P* value cut-off of <0.1. Second, given our overarching hypothesis that better quality or quantity of maternal care is protective against an adverse stress phenotype in offspring, we focused on CpG sites that exhibited the opposite direction of associations for the early social metrics vs. fGCMs. That is, our hypothesis proposes that a more favorable early social environment is associated with a better stress phenotype later in life. If this is the case, then a CpG site that is positively associated with maternal care (i.e., better quantity or quality of maternal care correlates with higher DNA methylation of that CpG site) should be inversely related to cortisol (i.e., higher DNA methylation of the CpG site should be associated with lower fGCMs). Thus, CpG sites of interest should theoretically be associated with maternal care and fGCMs in opposite directions in order to be potential markers of this relationship.

To interpret DMSs of interest, we copied an ~20 kb sequence of spotted hyena DNA centered on the CpG site of interest and saved the sequences as Fasta files. We first mapped each DNA sequence containing the CpG site of interest to the domestic cat (*Felis catus* Nov. 2017 [felCat9] Assembly) using the UCSC Genome Browser tool, the BLAST-like alignment tool (BLAT)[23] and identified nearest genes from multiple species alignment. Next, we mapped each hyena DNA sequence to the human genome (UCSC Human Dec. 2010 [GRCh38/hg38] Assembly) to again identify the nearest gene(s) and cross-checked the alignments with those from the more closely related cat genome. We used default BLAT parameters (see Supplementary Material) and retained the highest-scoring DNA sequence matches in cases where there was concordance between the multiple species alignment when querying hyena DNA against the cat and human reference genomes. We do not report the nearest human gene(s) when there was discordance between cat and human queries and when the BLAT score and % identical sequence matches were low (<500 BLAT score, <85% match).

**Reporting summary**. Further information on research design is available in the Nature Research Reporting Summary linked to this article.

## Data availability
The full data used in this paper are available at https://doi.org/10.5281/zenodo.4967924 [103]. Source data are provided with this paper. Sequence data for the spotted hyena putative glucocorticoid receptor promoter region are available at GenBank at https://www.ncbi.nlm.nih.gov/nuccore/MZ399371, accession number BankIt2471386 Crocuta MZ399371. Short read sequence data from the epigenome wide association study are available at the Sequence Read Archive at https://www.ncbi.nlm.nih.gov/bioproject/PRJNA740026, accession number PRJNA740026.

## Code availability
Source code is provided with this paper at https://doi.org/10.5281/zenodo.4967924 [103].

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

## Acknowledgements

We thank the Kenyan National Commission for Science, Technology, and Innovation, the Naboisho Conservancy, the Narok County Government, the Kenya Wildlife Service (KWS) and Brian Heath for permission to conduct research in the Mara ecosystem. We also thank the Mara Hyena Project field crew and the residents of the Mara ecosystem for their direct and indirect support of long-term field research in their backyard. We are indebted to all those who have contributed to long-term data and sample collection on the Mara Hyena Project, including undergraduate researchers, Karee Lesko, Shannon Carvey, and Kaycee Morra. We are also grateful to Dr. Elise Zipkin for suggestions and feedback regarding the statistical modeling, as well as Dr. Lu Tang and Patrick Bills for help processing the behavioral data. This work was supported by National Science Foundation Grants DEB1353110, OISE1556407, and IOS1755089 to KEH, and Doctoral Dissertation Improvement Grant from NSF (DDIG 1701384) to ZML. This work was also supported in part by funds from NSF Grant OIA 0939454 to the BEACON Center for the Study of Evolution in Action as well as Michigan Lifestage Environmental Exposures and Disease (M-LEEaD), NIEHS Core Center (P30 ES017885), as well as the UM NIEHS Institutional Training Grant T32 ES007062 to DCD. Dr. Perng is supported by the Center for Clinical and Translational Sciences Institute via KL2-TR002534.

## Author contributions

Conceptualization, Z.M.L.; data generation and curation, Z.M.L., J.R.G., J.W.T., T.M.M., M.A.S., L.S., R.G.C., K.R.P., C.L. M.O.P.; formal analysis, Z.M.L., R.G.C., W.P.; funding acquisition, Z.M.L., C.D.F., D.C.D., K.E.H.; investigation, Z.M.L.; methodology, Z.M.L., R.G.C., K.R.P., C.L., B.v.H., C.D.F., K.E.H., W.P.; resources, Z.M.L., C.D.F., D.C.D., K.E.H.; supervision, B.v.H., C.D.F., D.C.D., K.E.H., W.P.; visualization, Z.M.L. R.G.C.; writing – original draft, Z.M.L.; writing – review & editing, Z.M.L., J.R.G., J.W.T., T.M.M., M.A.S., M.O.P., L.S., R.G.C., K.R.P., C.L., B.v.H., C.D.F., D.C.D., K.E.H., W.P.

## Competing interests

The authors declare no competing interests.
