## [Peer Review File · Nature Communications]

REVIEWER COMMENTS

Reviewer #1 (Remarks to the Author):

This manuscript seeks to test the hypothesis that DNA methylation represents a mechanistic link between early social experience and a biomarker of stress later in life. The section evaluating whether unfavorable early social experiences affect later stress physiology is particularly clear and an important contribution to the field. However, the later sections examining global DNA methylation and genome-wide DNA methylation appear inconclusive and are difficult to interpret. Specifically:

1) There are substantial limitations to studying global DNA methylation, namely the absence of a clear mechanistic link and difficulty to interpret the consequences of small effects. Some previous studies have avoided these pitfalls with either large effects (e.g. cancer genomes) or experimental designs which provide a clear model (e.g. folate restricted diets in rodents). However, in the current study, it's not clear if the observed difference in whole genome methylation (a couple of percent) is biologically meaningful [is it enough to stimulate genome instability?] or how it relates to the predictor variables [are methyl groups so limited in the hyena diet that a good or bad year matters?]. The absence of this information makes it difficult to evaluate whether the reader would expect global DNA methylation to be a good biomarker and how to interpret the small effect sizes observed.

2) As the authors note near the end of the discussion, cell type variation is an important source of variation that must be accounted for in modeling DNA methylation levels (such as by estimating proportions of major cell types using blood smears).

Similarly, genetic covariance between samples is important and can dramatically effect results, especially when only considering a few individuals, and should be accounted for even if only from pedigree records or genotypes called from bisulfite converted DNA (i.e. bisSNP).

3) 29 samples is small for a genome-wide association study, especially given the biological noise in estimating DNA methylation and fGCs. I believe a power-analysis in the SI would be extremely useful. On the topic of power, it is fairly typical to exclude CpG sites with very low or high methylation (e.g. <10% or >90%), which may increase signal by reducing the multiple-testing burden.

Furthermore, the distribution shown in Figure S8b is depleted for low p-values and not nearly uniform, suggesting a permutation-based null and FDR correction may be a better fit than the typical Benjamini-Hochberg correction. I think this would be apparent in Figure S8a if the $y=x$ line was added as well, although there may be an enrichment over a permutation-based null.

On a related note, the combination of global and genome-wide DNA methylation levels opens the possibility to test what genomic contexts may be driving the global patterns observed. Are there systematic biases in the effect sizes for genomic contexts (CpG islands, gene introns, gene exons, promoter regions, GO categories, etc.) which are in the same direction as the global effects and could be

driving the overall patterns?

Minor points:

-Abstract: it's unclear what "DNA methylation biomarkers" means, especially considering that, at this point, there has been no mention of genome-wide DNA methylation. I would suggest rephrasing as "Finally, using bisulfite sequencing data, we identify XXX CpG sites".

-Introduction: It's mentioned that "few studies have measured natural variation..." but gives no mention of those that do. A few citations regarding where this has been done, and therefore what insights can be gained, may be helpful here. Similarly, "few have measured all three elements in the same population" would benefit from examples which do so which could be integrated into the explanation of why it is variable. Currently the only citation is to Weaver et al. 2014 which, while foundational, does not utilize a genome-wide approach.

-Methods:

It's mentioned that "the majority of CpG sites assessed via LUMA occur in gene bodies", but is that actually true or are gene bodies just strongly enriched? It would be helpful to extract the location of CCGG sequences and report what percent are contained in promoters, gene exons, gene introns, CpG islands, CpG shores, etc. to know which regions of the genome LUMA is weighted by.

Similarly, what was the coverage of the genome-wide DNA methylation data and the distributions for site-specific mean methylation and the genomic context of measured CpG sites?

What is the NCBI link/published citation for the hyena genome assembly used? If it hasn't been done already, it would be useful to have genes annotated via NCBI or ensemble. UCSC also has publicly available tools to identify CpG islands which are very informative for analyses of DNA methylation and not presented here.

What were the parameters used for blat, and how were multiple complete/best hits resolved? Was there a maximum distance for the nearest gene?

-Results:

What is the logic behind showing the uncorrected model in Table 1? It doesn't seem particularly useful given that, as the authors note, there are relevant corrections.

Following the EWAS analysis between DNAm and fGCMs, it appears as if only 15 sites were tested for associations with maternal care and rank. This type of partial analysis based on arbitrary "significant" sites reduces the ability to identify shared effects (Urbut et al. 2018, Nat Gen) and also limits the ability to test whether such effects are more likely than expect by chance. I would suggest doing the full EWAS for the other predictor variables and testing for overlap/enrichment and whether the subset of sites identified truly survive multiple hypothesis testing.

Discussion: Overall this was really well written and did a good job discussing the results and the weaknesses. I would just point out that the yellow-bellied marmot example is a species that is not

obligately social (unlike the other species mentioned) and therefore may be the exception that proves the rule.

-Candidate gene study: What is the window size and other features of the putative promoter for the glucocorticoid receptor? Does it overlap a CpG island or contain more CpG sites in other species? The relative lack of CpG sites and DNA methylation may be explained by the lack of a CpG island in the promoter of this species, the promoter was misidentified, or the bisulfite sequencing was off target. Given the surprising nature of this result, I think more detail may help a reader make sense of what happened or, if unable to disentangle possible causes, maybe it should be omitted to avoid unnecessary confusion.

-Figure 3: The DAG figures are useful to understand the study design, but would it be possible to integrate the results into the same figures such as by weighting the arrows based on the effect sizes and including an annotation for which were significant? Otherwise, upon a quick skim, it may appear as if these are the observed connections rather than all of the links tested.

Reviewer #2 (Remarks to the Author):

This paper was a pleasure to read and review. It describes a highly interesting set of studies with relevance to a broad range of disciplines. Of particular interest is the examination of the impacts of maternal behavior and social environment on epigenetic marks and corticosteroid levels in a complex social and ecological environment. As so much of both the epigenetic and stress literature relies on either rodents in highly deprived artificial environments or human studies which have to cope with high degrees of unobserved variability and reliance on self-report, the study should help resolve some long standing questions with regard to development and stress. I'm particularly struck by the fact that maternal behavior has such a limited effect on adult corticosteroid levels, while the richness of social contacts outside of the den appears to be more powerful. I think this may begin to explain why so many early life interventions in humans fail to produce substantial effects across the lifespan, and point the way to new developmental windows in which to intervene. The paper was well written and should be published with a commentary calling attention to it. I have only a few substantive concerns:

1. P-7. Description of the LUMA assay- states: "The majority of CpG sites assessed...occur in gene bodies ... as well as in non-coding regions of the genome..." As this description basically covers the entire genome, I'm not sure how meaningful the argument is. I think it might be better to simply cite some references that associate global DNA methylation with either health, epigenetic age or some equivalent variable.

2. Following the thought above. Would it be possible to examine epigenetic age in these samples? If so I think it would substantially enhance the paper.

3. For the EWAS analysis (and more generally), why was the domestic cat used when a draft genome of the both the striped and spotted hyenas exist (Yang et al 2020)? Even if these draft assemblies are too spotty to be useful in all cases, it should be noted they do exist and explained why they were not used.

Similarly, it would increase my confidence in the data if the approach of using the cat genome was validated using another well-annotated carnivore such as the domestic dog.

4. With regard to the glucocorticoid receptor data, is it possible the site examined was a pseudogene? If not, it might be useful to give a bit more context in terms of hyena endocrinology, which is relatively unique, and might explain why this species appears to have no plasticity at this gene, which seems to have plastic expression/epigenetic regulation in most other mammalian species thus far examined.

Yang C, Li F, Xiong Z, Koepfli KP, Ryder O, et al. 2020. A draft genome assembly of spotted hyena, *Crocuta crocuta*. *Sci Data* 7: 126

Reviewer #3 (Remarks to the Author):

Laubach et al.'s interesting manuscript entitled Associations of early social experience with offspring DNA methylation and later life stress phenotype represents an important investigation of the role of social interactions/connectedness on subsequent genetic (DNA methylation) and stress responses (fecal-based glucocorticoid metabolites).

The use of a novel model species ---the spotted hyena---in a naturalistic setting brings much needed authenticity to previous investigations in this research area that are based on laboratory bred animals raised in restricted laboratory environments. Further, the importance of social interactions in natural hyena behavior and the relevance of their social hierarchy on survival-related resources make this species ideal for the proposed research in the current study.

It's not surprising that the data depart slightly from earlier findings in related studies focusing on maternal behavior and subsequent hypothalamic-pituitary-adrenal (HPA) activity in developing offspring. For example, it was reported in Laubach's manuscript that observations of maternal care early in hyena offspring development seemed to be less impactful than reported in rats [e.g., by Michael Meaney's team's work on low-licking and high-licking maternal rats (e.g., see Caldji et al., 1998, Maternal care during infancy regulates the development of neural systems mediating the expression of fearfulness in the rat. *Proceedings of the National Academy of Sciences of the United States of America*, 95(9), 5335–5340)]. However, by focusing further in the developmental phase (i.e., Den Independence phase), social interactions were indeed found to be associated with fecal metabolites in a direction indicating that increased social interactions mitigated the glucocorticoid response. Still, Laubach et al.'s findings suggest that the maternal-offspring interactions may be more complex than previously observed in the rodent studies. For example, it is possible that more vulnerable offspring may be viewed to elicit more maternal responses than healthier animals when intra- and inter-litter differences are

considered. The limitations of collecting fecal samples at various times throughout the day were addressed by treating the time of collection as a covariate in the statistical analyses.

Looking forward, it would be interesting to also include DHEA assays when processing fecal samples, as considering DHEA/CORT ratios provide another index of adaptive stress responsivity since DHEA has been associated with emotional resilience, among other functions (e.g., see: Sripada RK, Marx CE, King AP, et al. DHEA enhances emotion regulation neurocircuits and modulates memory for emotional stimuli. *Neuropsychopharmacology*. 2013;38(9):1798-1807). The fact that the samples weren't collected after a stress challenge limits the information about responsivity but this is addressed in the manuscript. Looking back in the data, there may have been isolated times after a natural stressor that may be interesting to try to tease apart from the pooled baseline data.

Focusing on the findings that higher social connectedness during the maternal and later developmental phase affects global DNA methylation corroborate previous work, and enhances the robustness of these earlier findings with ecological relevant data in a unique species model.

Concerning the scoring of maternal behavior, it wasn't clear if proximity of the infant to the mother was viewed independently from the nursing and grooming behavioral categories. Was proximity scored when the animals were in proximity but NOT engaging in those behaviors? Was social play considered in the early behavioral observations?

The sophistication of the statistical approaches in this manuscript (e.g., linear mixed model) is impressive and appears to consider the most likely causal paths with the social connectedness-stress response scenario. With the data being drawn from a pool of observations/samples extending from 1988-2016, is it necessary to describe other studies published from these data? I see that one study apparently published from this data set is mentioned in the current manuscript (i.e., Laubach et al., 2019, *Mol Ecol*), but doesn't explicitly convey that the findings were in the same group of animals/data.

To succinctly address the prescribed questions highlighted for reviewers, the findings described in the current manuscript are indeed noteworthy as they provide a naturalistic, ethologically-relevant approach to genetic and endocrine markers of emotional resilience and stress. This approach qualifies specific aspects of past findings with rodent models. The findings support the claims in the study and, although the methodology has limitations in areas (e.g., not being able to control time of fecal sample collections to control for circadian effects), the authors cleverly compensated for these variations with statistical techniques. The methods are clearly described in the manuscript and supplemental materials.

Reviewer #4 (Remarks to the Author):

In the manuscript entitled "Associations of early social experience with offspring DNA methylation and later life stress phenotype" the authors chose wild spotted hyenas (*Crocuta crocuta*) as model species aiming to decipher the impact of early life experiences on faecal glucocorticoid metabolite levels and DNA methylation. The authors also test the hypothesis that DNA methylation may act as physiological

mediator among early life experience and stress later in life. The study was inspired by former studies in laboratory model species under controlled experimental set-ups.

Early experiences were measured by observation of maternal care which were specified as grooming, nursing and close proximity to the offspring's mother during two chosen time windows in early development "communal den (CD) phase" and the subsequent "den-independent (DI) phase".

Unfortunately, it remains unclear why these specific windows were chosen, as well as their lengths in months. The biological meaning of splitting "maternal care" into the three variables: grooming/licking, nursing and close proximity needs further explanation. Figure 6 shows that only close proximity of offspring to the mother has an effect on global methylation, but neither grooming nor nursing. The biological meaning of this but not the others remain unexplained in the discussion, where the authors refer to this result as maternal care, instead of to close proximity. "We also found that more maternal care in the first year of life, which roughly corresponds with the CD period of development,". The same strategy of gathering the results which were before specified holds true for measures of the CD period, stated in the second part of this sentence "...and greater social connectedness during the DI phase were each associated with higher global DNA methylation, a presumed indicator of genomic stability and overall health.", while figure 7 shows only a slight effect for the "degree" of interactions on the global methylation ratio, but not the others. Why were these variables separated into more specific variables, but explained as one? What is the biological relevance of these specifications?

Furthermore, the specific maternal care observatory data were measured in 1 to 29 repeated observations of mother-offspring interactions. These variable estimated values were subsequently reduced to a single value (by calculating the distance to the mean of each measurement), erasing its variation, and thus reducing the complexity of the data. This approach has been shown to be problematic as this strategy leads to a decrease in robustness of the model these values are used in (<https://academic.oup.com/beheco/article/28/4/948/3059669>).

Also it is unclear why certain covariates were used in respective models e.g. one includes social rank, but not the others. Please explain how they were chosen.

The offsprings' sex seems critical, because females are philopatric while males disperse. As a result it may lead to differences in early behaviour. Additionally, puberty in males occurs earlier (after 18-24 months) than in female spotted hyenas (after 24-30 months), which may also lead to differences in early behaviour, and has been shown to change the overall DNA methylation in human. Please provide the number of males and females in the manuscript.

A comparison of a wild mammal to non-model species seems like the attempt to simplify a highly complex system within an uncontrolled experimental set-up. This simplification is critical.

The authors are partly aware of this as they state within the discussion "..., in contrast to controlled rodent experiments, wild hyenas are subject to a multitude of stressors over development, which may hamper our ability to isolate the specific effect of maternal care on stress phenotype. Second, compared to maternal separation common in many experimental studies, natural variation in maternal care is far more subtle and may require a more sensitive measure of physiological stress than average fGCMs, which is a summary baseline stress indicator 51 subject to unmeasured environmental factors and the animal's condition when the sample is collected 52."

The measurement of faecal glucocorticoid metabolite is a valid method. But stress levels are prone to outer environmental changes e.g. predatory attacks and human interactions increase stress levels. Accounting for these events is often difficult due to limited time windows of observation. "Behavioral data were collected daily between 0530 – 0900 h and 1700 – 2000 h." Therefore replicates of fGCMs for each animal are important as the period and day of sampling is critical. Please state if they were applied.

Global methylation analysis using LUMA only provides a methylation ratio over the entire genome. It does not provide the data for functional interpretation. The authors state "The majority of CpG sites assessed via LUMA occur in gene bodies, where they may function in transcription regulation and alternative splicing 27, as well as in non-coding regions of the genome, where they may repress repetitive elements 28 and enhance chromosome stability 29. Therefore, we assumed that lower than average %CCGG methylation is likely disadvantageous to health." This assumption is very farfetched, because LUMA does not provide the details to distinguish between functional important regions and mostly excludes promoters, which are thought to be the main regulatory regions. Therefore methylation changes are difficult to interpret and should be handled with caution. Using LUMA is cost-efficient and does not need a reference genome, which would be two valid reasons to use it. However the usage of three different approaches with different samples numbers: global DNA methylation (n = 186), genome-wide DNA methylation (n = 29), and candidate gene DNA methylation (n = 96) without explanation, is confusing. Please clarify why you chose this strategy.

In order to assess the biological meaning of the established data, please clarify the numbers of animals, mothers, sex of offspring, and siblings, and the pairwise interactions observed. The number of samples collected for CD and DI time windows, stating why these were chosen, the sex, the cofounding factors that can lead to an increase in stress, the robustness of the metabolites measured. Last but not least, providing the data and code used is essential to understand and repeat the analysis.

POINT-BY-POINT RESPONSE TO REVIEWER COMMENTS

We want to begin by thanking all of the reviewers for their thoughtful and timely review of our manuscript. We have carefully considered the reviewers' comments below and sought counsel from multiple statisticians (n=2), bioinformaticians (n=1), genetic/genomic epidemiologists (n=2) and evolutionary biologists (n=1) in order to address the issues identified by the reviewers, particularly in relation to the global and genome-wide DNA methylation analyses. We addressed specific comments below by carefully weighing the consensus opinions of the reviewers, the experts with whom we consulted, and drawing on the specializations represented in our author list. In pooling expertise from multiple sources, we acknowledge that multiple approaches are employed in studies like ours and that this is an area of rapid methodological development (e.g. causal inference, statistical analyses of genomics etc.), and so the methodologies employed herein represent what we collectively determined to be the most appropriate options. We feel that the changes made based on the reviewers' comments have improved this work, and we respond to specific concerns below.

Reviewer #1:

1. **Comment:** This manuscript seeks to test the hypothesis that DNA methylation represents a mechanistic link between early social experience and a biomarker of stress later in life. The section evaluating whether unfavorable early social experiences affect later stress physiology is particularly clear and an important contribution to the field. However, the later sections examining global DNA methylation and genome-wide DNA methylation appear inconclusive and are difficult to interpret. Specifically (see following comments):

Response: We are grateful to the reviewer for their careful review of this manuscript, and we are delighted that they find value in this work as a contribution to the field.

2. **Comment:** There are substantial limitations to studying global DNA methylation, namely the absence of a clear mechanistic link and difficulty to interpret the consequences of small effects. Some previous studies have avoided these pitfalls with either large effects (e.g. cancer genomes) or experimental designs which provide a clear model (e.g. folate restricted diets in rodents). However, in the current study, it's not clear if the observed difference in whole genome methylation (a couple of percent) is biologically meaningful [is it enough to stimulate genome instability?] or how it relates to the predictor variables [are methyl groups so limited in the hyena diet that a good or bad year matters?].

Response: We agree that there are limitations to studying global DNA methylation, and we have noted such limitations in the discussion. However, we chose this epigenetic mark for logistical and scientific reasons. Logistically, at the time that the laboratory work was carried out, the hyena genome (which only recently became publicly available in 2020) had not yet been sequenced. Use of the LUMA assay does not require a sequenced genome, and thus, was a viable and cost-effective approach to assessing methylation of a ubiquitously present DNA sequence in a novel wild animal model. Scientifically, there is an established literature on the utility and relevance of global DNA methylation as a developmentally labile biomarker (Fraga et al., 2005) that is both associated with early life experiences (c.f. Crudo et al., 2012; Doherty, Forster, & Roth, 2016; Noguera & Velando, 2019) and later life health outcomes (c.f. Dong et al., 2017; Smolarek et al., 2010; Wong et al., 2014; Zinellu et al., 2017). To address this comment, we have updated the Introduction and Discussion to provide biological context, and we have clarified the interpretation of our global DNA methylation results.

Introduction:

Pgs 3-4, Lines 35-38:

Subsequent work in rodents demonstrated that maternal separation correlated with widespread differences in brain tissue DNA methylation, not only at single promoter regions, but also in the form of lower global DNA methylation (Anier et al., 2014).

Methods:

Pg 8, Lines 124-130:

Coupled with evidence that global DNA methylation is responsive to early-life environmental factors (Crudo et al., 2012; Doherty et al., 2016; Noguera & Velando, 2019), and the fact that lower global DNA methylation is associated with a range of adverse health outcomes in general-risk human populations (e.g., shorter telomere length as a metric of accelerated aging (Dong et al., 2017; Wong et al., 2014); hypertension (Smolarek et al., 2010); and chronic obstructive pulmonary disease (Zinellu et al., 2017)), we interpret average %CCGG methylation as a relevant biomarker of developmental plasticity where lower global methylation is disadvantageous to health.

Discussion:

Pgs 34-35, Lines 574-585:

Our findings that less maternal care during the first year of life and less social connectedness during the DI phase of development are associated with lower global DNA methylation suggest that inadequate early social experiences may be risk factors for suboptimal health later in life. Previous research in humans has shown that lower global DNA methylation corresponds with adverse health outcomes related to cardiovascular disease (Smolarek et al., 2010), pulmonary disease (Zinellu et al., 2017), shorter telomere length (a marker of accelerated aging) in both adolescents (Dong et al., 2017) and adults (Wong et al., 2014), as well as cancer (Woo & Kim, 2012). It is worth noting that higher global DNA methylation has also been associated with some cardiovascular diseases (Sharma et al., 2008). Thus, deviations in either direction, both lower or higher DNA methylation, may be a marker of aberrant physiology and should be interpreted with caution.

Pg 35, Lines 589-977:

We found no evidence of mediation by %CCGG methylation. This could be due to low power given the relatively small sample sizes, and/or related to the fact that %CCGG methylation averaged over the entire genome may be too broad a metric to isolate important regulatory pathways involved in the stress response. For example, we previously found that maternal prenatal socioeconomic status was associated differential methylation at specific regions of the genome but not with global DNA methylation in humans (Laubach, Perng, et al., 2019). Therefore, we complemented our global DNA methylation analysis in hyenas with mERRBS (Meissner et al., 2005), a high throughput sequencing technology that assays single nucleotide DNA methylation resolution at a genome wide scale.

- 3. Comment:** The absence of this information makes it difficult to evaluate whether the reader would expect global DNA methylation to be a good biomarker and how to interpret the small effect sizes observed.

Response: Previous studies have shown larger effect sizes when assessing global DNA methylation in relation to cancer, and as a consequence of inadequate methyl donor intake. However, these scenarios represent extreme cases of genomic instability and nutrient restriction, respectively, and may not be relevant to the majority of individuals in a population. The goal of this study was to understand whether natural variation in early social experience is associated with naturally-occurring differences in global DNA methylation, especially given earlier findings that the hyenas' maternal social rank and food

availability correspond with similar magnitudes of effect sizes when %CCGG methylation is the outcome (Laubach *et al.* 2019. *Mol. Eco.* 28: 3799-3812). The effect sizes we report in this analysis ($\beta = 1\%$ to 2%) are modest but represent differences detectable across many CCGG sites (~2.1 million) throughout the genome and thus, likely have implications for health and potentially fitness. Moreover, the differences we observed in this analysis are comparable or larger than those reported in observational studies of healthy humans. To list a few:

- In an analysis of 384 healthy Japanese women, Ono *et al.* detected differences %CCGG methylation of $\leq 1\%$ with respect to known determinants of global DNA methylation, including **smoking habits** (0.5% difference in %CCGG methylation for current vs. never smoker), **age** (0.1% to 0.5% difference across a range of age from 40 to ≥ 70 years), and **dietary habits** ($< 1\%$ difference across quartiles of folate; vitamins B2, B6, and B12; and alcohol intake), (Ono *et al.* 2012. *Can. Sci.* 103: 2159-2164)
- Several observational studies in healthy individuals have assessed global DNA methylation (albeit, these studies use LINE-1 DNA methylation as a proxy for global DNA methylation) in relation to various environmental exposures, as well as a determinant of subsequent health outcomes. These analyses report even smaller differences in global DNA methylation. In a study of 568 healthy schoolchildren in Bogotá, Colombia, the authors detected small differences in global DNA methylation with respect to **sex** (0.21% higher in boys), **low-grade inflammation** (0.12% with respect to CRP ≥ 1.0 mg/L vs. < 1.0), **serum vitamin A concentrations** – a regulator of oncogene activity (0.07% to 0.19% lower methylation, respectively, for adequate and excessive vs. inadequate serum levels), and **socioeconomic status** (0.29% difference for highest vs. lower stratum in the sample) (Perng *et al.* 2012. *Epigenetics.* 7: 1133-1141). Such differences in DNA methylation, though small, were subsequently linked to clinically-relevant rates of adiposity gain during 2.5 years of follow-up (Perng *et al.* 2013. *PLoS One.* 8: 1-7)

We also acknowledge that larger differences in %CCGG methylation have been reported in studies that do not focus on cancer – e.g., ~6% lower DNA methylation in brain tissue of wild polar bears in response to mercury exposure (Pilsner *et al.* 2010. *Mol. Eco.* 19: 307-314) and differences of 7% to 8% in tail tissue of *Agouti* mice with respect to varying levels of BPA exposure (Anderson *et al.* 2012. *Env. Mol. Mut.* 53: 334-342). However, these studies assayed DNA methylation in target organ tissue, which one would expect to exhibit larger effect sizes than a systemic tissue like peripheral leukocytes. Finally, note that recent expert consensus coalesces around the opinion that even small effect sizes are meaningful, particularly in studies that test developmental origins hypotheses (Breton *et al.* 2017. *Env. Health Persp.* 125: 511-526). To address the reviewer's comment, we have made the following additions to the text:

Discussion:

Pg 35, Lines 583-585:

Finally, the effect sizes for %CCGG methylation we observed are modest but biologically relevant, as they are comparable in magnitude to dietary (Ono *et al.*, 2012) and toxicant (Basu *et al.*, 2013) exposures reported in the existing literature.

4. **Comment:** As the authors note near the end of the discussion, cell type variation is an important source of variation that must be accounted for in modeling DNA methylation levels (such as by estimating proportions of major cell types using blood smears).

Response: Unfortunately, we do not have information on the cell type composition, which we discussed as a potential limitation. However, the absence of these data does not affect interpretation of our results given that our goal was to estimate the total effect of each explanatory variable on

%CCGG methylation. Cellular composition is likely on the causal pathway given that social stressors have been identified as determinants of cell type composition (Engler *et al.* 2004. *J. Neuroimm.* 148: 106-115) which in turn, is a known determinant of DNA methylation. In such a scenario, adjustment for cell type composition would introduce rather than reduce bias into the estimate of interest (Ananth & Schisterman. 2017. *Am. J. Obst. Gyn.* 217: 167-175; Schisterman *et al.* 2009. *Epid.* 20: 488-495). To address this comment and provide further clarity, we have modified the following text in the Discussion section:

Discussion:

Pg 39, Lines 682-687:

Fourth, we used archived DNA from whole blood that lacked information on cell type composition, and therefore we were not able to control for cell type heterogeneity as a source of variation in DNA methylation. However, cell type composition may be influenced by factors like social stress (Engler *et al.*, 2004), and therefore could be on the causal pathway between the explanatory variables of interest and DNA methylation. Accordingly, adjusting for such a variable in the analysis would introduce rather than reduce bias (Ananth & Schisterman, 2017; Schisterman *et al.*, 2009).

5. **Comment:** Similarly, genetic covariance between samples is important and can dramatically effect results, especially when only considering a few individuals, and should be accounted for even if only from pedigree records or genotypes called from bisulfite converted DNA (i.e. bisSNP).

Response: In response to this comment, we ran microsatellite assays to determine paternity, used these data to construct a pedigree and a covariance matrix of relatedness, and included the matrix in our EWAS models to control for genetic correlations. These updates are described in text (Methods: Pages 17-18, Lines 299-301; SI: Pages 16-17, Lines 301-307). For analyses in which global (%CCGG) DNA methylation was the outcome, we included a random intercept for maternal ID to account for familial clustering due to shared genetics. We have noted these changes in the main and supporting texts.

6. **Comment:** 29 samples is small for a genome-wide association study, especially given the biological noise in estimating DNA methylation and fGCs. I believe a power-analysis in the SI would be extremely useful.

Response: While our sample size is small (which we noted as a limitation, Conclusions: Page 39, Lines 680-682), it is comparable to other recent genome-wide DNA methylation studies in wild animals (c.f. Taff *et al.* 2019. *Mol. Eco.* 28: 3722-3737; Uren Webster *et al.* 2018. *Epigenetics.* 13: 1191-1207; which have sample sizes of $n = 24$). Further, the small sample was likely not prohibitive given that we were able to detect meaningful differences after false discovery rate correction. Of note, we used strict sample selection criteria to reduce noise in the data (Methods: Pages 16-17, Lines 278-282), thereby reducing the need for covariate adjustment and enhancing statistical power.

Regarding the suggestion to include a power analysis in the Supplemental Material: the EWAS portion of this analysis was exploratory, so we did not conduct *a priori* power analysis. Additionally, we do not feel it is appropriate to conduct *post hoc* power analysis given that we were able to detect significant effects after false discovery rate corrections, and because *post hoc* power analyses are prone to type 2 error (c.f. Bababekov *et al.* 2018. *Ann. Surg.* 267: 621-622, and a less-technical piece written by Gelman: http://www.stat.columbia.edu/~gelman/research/unpublished/power_surgery.pdf).

7. **Comment:** On the topic of power, it is fairly typical to exclude CpG sites with very low or high methylation (e.g. <10% or >90%), which may increase signal by reducing the multiple-testing burden.

Response: Thank you for the suggestion. We are aware of research groups who implement this initial data processing step (Lea *et al.* 2016. *Mol. Eco.* 25: 1681-1696). Following this precedent, we have removed 5% of CpG sites which exhibited the lowest variation in DNA methylation, and those that had <10% DNA methylation or >90% DNA methylation in order to reduce the burden of false positives. We then conducted the EWAS using this filtered data set and have made changes describing the data filtering and new analysis in the main and supporting text (Methods: Page 17, Lines 286-288; SI: Page 16, Lines 292-300; SI, Page 23, Lines 394-396).

8. **Comment:** Furthermore, the distribution shown in Figure S8b is depleted for low p-values and not nearly uniform, suggesting a permutation-based null and FDR correction may be a better fit than the typical Benjamini-Hochberg correction. I think this would be apparent in Figure S8a if the $y=x$ line was added as well, although there may be an enrichment over a permutation-based null.

Response: Thank you for this insightful point. After additional consultation with experts in genomics and statistics, we agree that it is appropriate to control for inflation and bias as indicated by our original P-value distribution and QQ-plot. We used the Bioconductor package, 'bacon' to correct estimates and associated P-values for inflation and bias. We describe these changes in the main and supporting text, and we have provided additional supplemental figures that highlight improvements after correcting for bias and inflation.

Methods:

Pg 18, Lines 301-305:

After running the EWAS, we corrected estimates of association and P-values for inflation and bias due to unmeasured confounding using the Bioconductor package, 'bacon' in order to reduce false positive associations (van Iterson *et al.*, 2017). We accounted for multiple comparisons using a Benjamini-Hochberg false discovery rate (FDR) of 5% on the inflation and bias corrected P-values (Hochberg & Benjamini, 1990).

Supporting material:

Pg 24 (SI), Lines 403-408:

We graphed the P-value distribution (**Supporting Figure 9, a**) and a Q-Q plot of the expected vs. the observed P-values (**Supporting Figure 9, b**) from our uncorrected EWAS. These plots indicated the presence of test statistic bias and inflation, likely due to unmeasured confounding, so we used the package 'bacon' to improve estimation of the empirical null distribution (van Iterson *et al.*, 2017). After applying the 'bacon' correction we no longer observed depletion of low P-values (**Supporting Figure 9**).

Supporting Figure 9. Diagnostic plots for stress related EWAS. **a)** A frequency histogram showing the distribution of uncorrected P-values (gray bars) and bacon corrected P-values (blue bars) generated from the EWAS. **b)** QQ-plots showing the observed $-\log_{10} P$ -values versus the expected $-\log_{10} P$ -values for the uncorrected and bacon corrected P-values. A frequency histogram showing a relatively uniform distribution of P-values generated from the EWAS.

9. **Comment:** On a related note, the combination of global and genome-wide DNA methylation levels opens the possibility to test what genomic contexts may be driving the global patterns observed. Are there systematic biases in the effect sizes for genomic contexts (CpG islands, gene introns, gene exons, promoter regions, GO categories, etc.) which are in the same direction as the global effects and could be driving the overall patterns?

Response: We thank the reviewer for this suggestion. Using the recently published spotted hyena genome (Yang *et al.* 2020. *Sci. Data.* 7: 1-10) we have annotated the 'CCGG' motif targeted by our global (LUMA) DNA methylation assay and the CpG sites from our EWAS. We used the UCSC genome browser tool, 'cpg_lh' to annotate 'CCGG' motifs in CpG islands, CpG shores, CpG shelves, and inter-CpG regions (c.f., Supporting Figure 8; Methods: Pages 7-8, Lines 119-130; Results: Page 26, Lines 448-455; SI: Page 6, Lines 101-109; SI: Page 18, Lines 330-337). Similarly, we annotated each differentially methylated site from our EWAS to each of these respective regions (c.f., Supporting Figure 11). We describe these methods and results and discuss their biological meanings in respective sections of the main text and supporting material. However, the present analysis is not an ideal setting in which to formally compare global vs genome-wide results for a few reasons. First, the overlap in the global and genome-wide DNA methylation samples includes only 12 animals, therefore a direct comparison between data sets is limited. Second, the purpose of the EWAS analysis was exploratory, and to identify potential functional pathways that can be assessed more rigorously in future analyses with larger sample sizes. At this point, we hesitate to add additional analyses on top of multiple existing analyses given the limits of article lengths for the journal and because of the already-dense nature of the paper. However, we mention these ideas in our revised manuscript as a suggestion for future research.

Supporting Figure 8. The 'CCGG' motif, which is targeted by the LUMinometric Methylation Assay (LUMA), annotated in the spotted hyena genome. a) Annotation categories are based on CpG density using the UCSC genome tool 'cpg_lh'. CpG shores include the region of the genome 2kb up and downstream of the CpG island start and end boundaries. CpG shelves include the region of the genome 2kb up and downstream of the CpG shore start and end boundaries. The interCGI region includes the remaining hyena genome. b) Genic annotations are based on the draft hyena genome and the general feature format (GFF) file. Annotated counts add up to more than 2,191,145 unique 'CCGG' motifs due to this 4-bp motif overlapping annotation boundaries.

Supporting Figure 11. Annotated counts of CpG sites identified from the stress hormone EWAS in spotted hyenas. **a)** Counts of differentially methylated sites (red bars = hypermethylated; blue bars = hypomethylated) annotated based on CpG density from the spotted hyena genome using the UCSC genome tool 'cpg_lh'. CpG shores include the region of the genome 2kb up or downstream of the CpG island start and end boundaries. CpG shelves include the region of the genome 2kb up or downstream of the CpG shore start and end boundaries. The interCGI region includes the remaining hyena genome. **b)** Counts of differentially methylated sites summarized by genic annotations, which are based on the draft hyena genome and the general feature format (GFF) file. In both panels a) and b), significant DMSs are determined after P-values are bias and inflation corrected and controlled for multiple comparison using Benjamini-Hochberg false discovery rate (FDR) of 5%. **c)** Annotated counts of all CpG loci tested in the stress hormone EWAS based on the CpG density annotation. **d)** Annotated counts of all CpG loci tested in the stress hormone EWAS based on genic annotation.

Minor points:

10. **Comment:** -Abstract: it's unclear what "DNA methylation biomarkers" means, especially considering that, at this point, there has been no mention of genome-wide DNA methylation. I would suggest rephrasing as "Finally, using bisulfite sequencing data, we identify XXX CpG sites".

Response: Done

Abstract:

Pg 2, Lines 8-10: Using bisulfite sequencing data, we also identified differential DNA methylation near 5 genes involved in inflammation, immune response, and aging that may link maternal care and stress phenotype.

11. **Comment:** Introduction: It's mentioned that "few studies have measured natural variation..." but gives no mention of those that do. A few citations regarding where this has been done, and therefore what insights can be gained, may be helpful here. Similarly, "few have measured all three elements in the same population" would benefit from examples which do so which could be integrated into the explanation of why it is variable. Currently the only citation is to Weaver et al. 2014 which, while foundational, does not utilize a genome-wide approach.

Response: We have now cited relevant studies in the Introduction, as suggested:

Introduction:

Pg 4, Lines 45-57:

First, there is a lack of studies that measure natural variation in the quantity and quality of early social experience, as much of this research includes experimental studies involving maternal separation and peer isolation, which are routine procedures in studies of captive primates and rodents^{14,15}. While informative, studies of extreme deprivation do not capture the range of normative social experiences that are relevant to development^{3,16,17}. Second, although studies have examined the relationships among social experiences, DNA methylation, and stress phenotype in piecemeal, few have measured all three components in the same population^{10,18-20}. Doing so is necessary to explicitly test the hypothesis that DNA methylation represents a mechanistic link between early social experience and future stress phenotype. Third, few studies explicitly consider that variation in the type and/or timing of social experiences may affect DNA methylation²¹ and stress phenotypes²², particularly in wild animal populations where development is subject to environmental fluctuation and pressures of natural selection.

12. **Comment:** Methods: It's mentioned that "the majority of CpG sites assessed via LUMA occur in gene bodies", but is that actually true or are gene bodies just strongly enriched? It would be helpful to extract the location of CCGG sequences and report what percent are contained in promoters, gene exons, gene introns, CpG islands, CpG shores, etc. to know which regions of the genome LUMA is weighted by.

Response: Thank you for noting this point requiring clarification. As suggested, we have annotated the 'CCGG' motif in the hyena genome and addresses this point in response to the reviewer's previous comment #9.

13. **Comment:** Similarly, what was the coverage of the genome-wide DNA methylation data and the distributions for site-specific mean methylation and the genomic context of measured CpG sites?

Response: Thank you, we have added new figures and text to provide added genomic context, (c.f. Supporting Figure 11, and additions to the methods, and results.)

Methods:

Pg 8, Line 139:

We filtered read counts to a minimum of 10x coverage.

Results:

Pg 26, Lines 448-455:

Based on the CpG density annotation, we observed 62% of CpG sites from our EWAS occurred in CpG islands and CpG shores, while among both hyper- and hypomethylated DMSs 72% of sites were in CpG islands and shores (**Figure S11, a, c**). According to our genic annotation, 38% of CpG sites from our EWAS occurred in either promoters, exons or introns, and 39% of DMSs were found in the same genic annotations (**Figure S11, b, d**). There were equal total numbers of hyper- and hypomethylated DMSs, but in both annotations more hypermethylated DMSs were found near CpG islands and gene bodies than in intergenic regions.

14. **Comment:** What is the NCBI link/published citation for the hyena genome assembly used? If it hasn't been done already, it would be useful to have genes annotated via NCBI or ensemble. UCSC also has publicly available tools to identify CpG islands which are very informative for analyses of DNA methylation and not presented here.

Response: The DNA methylation calling for the EWAS used the draft spotted hyena genome (at that time unpublished). We have now added the appropriate citation. We also address the identification of CpG islands in response to comment #9.

Methods:

Pg 8, Lines 137-139:

After DNA sequencing, we ran a standard bioinformatic pipeline to clean, align, and call DNA methylation reads using the draft spotted hyena genome (Yang et al., 2020).

15. **Comment:** What were the parameters used for blat, and how were multiple complete/best hits resolved? Was there a maximum distance for the nearest gene?

Response: We have added text clarifying the criteria we used for BLAT. We have also noted the approximate distance of each DMS based on the multiple species alignment from our BLAT query in Table 2.

Methods:

Pg 20, Lines 345-350:

We used default BLAT parameters (see Supporting Materials) and retained the highest scoring DNA sequence matches in cases where there was concordance between the multiple species alignment when querying the cat and human reference genomes. We do not report the nearest human gene when there was discordance between the cat and human queries and when the BLAT score and % identical sequence matches were low (<500 BLAT score, <85% match).

Supporting material:

Pg 17 (SI), Lines 309-317:

Among CpG sites that were differentially methylated in our EWAS, we used the web-based the BLAST-like alignment tool (BLAT) (Kent et al., 2002) tool from UCSC Genome Browser to map the region surrounding each DMS to the domestic cat (*Felis catus* Nov. 2017 [felCat9] Assembly) and human (UCSC Human Dec. 2010 [GRCh38/hg38] Assembly) genomes. Using the default settings, BLAT searches whole reference genomes using an 11-mer tile size and a step size of 5 to identify homologous regions between the query sequence of DNA and the reference genome (Kent et al., 2002). We report hits in which the hyena DNA sequence queried against cat and human genomes show agreement, and where the top BLAT score for matched alignments is > 500 and the percent identical match are > 85%.

16. **Comment:** Results: What is the logic behind showing the uncorrected model in Table 1? It doesn't seem particularly useful given that, as the authors note, there are relevant corrections.

Response: We show unadjusted associations in **Table 1** to present raw associations in the data, upon which the reader may assess the impact of subsequent adjustment for covariates. Such an approach is used in epidemiologic analyses as a valuable standalone approach for understanding crude associations (Conroy & Murray. 2020. *Brit. J. Canc.* 123:1351-1352), as well as in the context of multivariable analyses (e.g., Laubach *et al.* 2021. *Proc. Royal Soc. B.* 288: 2020815) to promote transparency in statistical analyses.

17. **Comment:** Following the EWAS analysis between DNAm and fGCMs, it appears as if only 15 sites were tested for associations with maternal care and rank. This type of partial analysis based on arbitrary "significant" sites reduces the ability to identify shared effects (Urbut *et al.* 2018, *Nat Gen*) and also limits the ability to test whether such effects are more likely than expect by chance. I would suggest doing the full EWAS for the other predictor variables and testing for overlap/enrichment and whether the subset of sites identified truly survive multiple hypothesis testing.

Response: While carrying out full EWAS on predictor variables (of which there are several) and fGCMs is certainly an option, this approach may be particularly prone to type 1 error for this study. This is because we have multiple early life social experience variables, so we would need to conduct multiple EWAS in addition to the stress related EWAS in order to be able to look for overlapping DMSs. Given the already small sample size for this portion of the analysis, we took the more efficient meet-in-the-middle approach of conducting a single EWAS with FDR correction to identify determinants of the outcome, and then assessing for associations with a key early life variable based on associations detected in the present analysis. We have used this approach in other analyses of high-dimensional data, including an epigenome-wide association study of prenatal SES and DNA methylation at three postpartum time-points (Laubach *et al.* 2019. *Epigenomics.* 11912: 1413-1427), and a metabolomics analysis that sought to identify novel markers of the relationship between sugar-sweetened beverage intake and hypertension (Perng *et al.* 2019. *Metabolites.* 9: 100).

18. **Comment:** Discussion: Overall this was really well written and did a good job discussing the results and the weaknesses. I would just point out that the yellow-bellied marmot example is a species that is not obligately social (unlike the other species mentioned) and therefore may be the exception that proves the rule.

Response: We have changed the text to make this clarifying point.

Discussion:

Pg 32, Lines 525-528:

For instance, a study on wild yellow-bellied marmots found no association of social network metrics with fGCMs measured in a mixed-age, mixed-sex population (Wey & Blumstein, 2012), although it should be noted that this species is not obligately social like those in other above-mentioned studies.

19. **Comment:** Candidate gene study: What is the window size and other features of the putative promoter for the glucocorticoid receptor? Does it overlap a CpG island or contain more CpG sites in other species? The relative lack of CpG sites and DNA methylation may be explained by the lack of a CpG island in the promoter of this species, the promoter was misidentified, or the bisulfite sequencing was off target. Given the surprising nature of this result, I think more detail may help a reader make sense of what happened or, if unable to disentangle possible causes, maybe it should be omitted to avoid unnecessary confusion.

Response: We have added information about the size and the %GC content of the putative spotted hyena GR promoter in the figure legend of **Supporting Figure 5**. In addition, this figure delineates important features including the location of potential NGFI-A transcription factor binding sites as well as consensus sequence based on alignments human and rodent DNA sequences. While we think these null results from hyenas are important to disseminate given that no prior study done this and because it contributes valuable information on the hyena GR promoter region, we are open to removing this finding if the reviewers and editors feel it is appropriate.

Supporting material:

Putative spotted hyena GR Promoter

```
5' - gggttctgctttgcaacttctctcccggggagagcgcgacggcggcgg
cggctgcagacggggcCGcccagacgatgcgggcggggggaacctgccgg
caocgcactacccccgagtcgagagtagtacgcaccgaccccctcctct
ctcctccccctccctcagcctccccagagggcgtgtctggtgtccggccc
cgagcagggccgagacgctgcgccaccgcttcccttgcaaccctcgtagcc
ccgtgctaagtgaacacagttcgcgcaactccgctccgagggcggcggcg
gacctcctccgcccggcctgctggggccgcccggcctccccccacc
cccacccCGccccagacagcccgtgtcaccgcaggggctgacggcg
ccggtcgcgagggactgggccaagctccggagtgggtcgggagtcgcgcc
gctgggctggggCGgaaggaggcagcagggagagaaactagagaaactc
ggcttccctcccaagctcgccccggagagaccaggtcggcctccagccg
cacctctcccttttcccggagggtggggggg - 3'
```

Pg 11 (SI), Lines 190-199:

Supporting Figure 5. Spotted hyena putative GR promoter sequence, which has been trimmed to match the aligned sequences published in Oberlander et al. 2008, McGowan et al. 2009, and Perroud et al. 2011. This sequence is 580 bp and has a 73.1% GC content. Underlined is the inferred consensus region (based on alignment) for human exon 1_F, and red, italicized is the inferred consensus region (based on alignment) for rat exon 1₇. GC boxes, the putative binding site for transcription factor NGFI-A, are **bold** and contained CpG sites capitalized. The yellow highlighted GC box is the potential NGFI-A transcription binding site homologous to the region in rats that was shown to be differentially methylated with respect to maternal licking and grooming (c.f. Weaver et al. 2004).

20. **Comment:** Figure 3 - The DAG figures are useful to understand the study design, but would it be possible to integrate the results into the same figures such as by weighting the arrows based on the effect sizes and including an annotation for which were significant? Otherwise, upon a quick skim, it may appear as if these are the observed connections rather than all of the links tested.

Response: As discussed in our recent publication (Laubach et al. 2021. *Proc. Royal Soc. B.* 288: 2020815), the purpose of DAGs is to depict the temporal and causal relationships among the variables

of interest. These graphs are conceptual and based on a priori knowledge. To clarify, we have updated all figure legends to reflect this (c.f. the updated legend for Figure 1).

Figure 1. Directed Acyclic Graph (DAG) for Part 1 of the analysis showing the hypothesized relationships between early-life social experience and adult fecal Glucocorticoid Metabolites (fGCMs). Covariates were included based on *a priori* biological knowledge and bivariate analysis (see Supporting Tables S2-S5). Precision covariates (gray text) were assessed when the fecal sample was collected. Variable positions over the timeline roughly corresponded with the timing of assessment.

Reviewer #2

1. **Comment:** This paper was a pleasure to read and review. It describes a highly interesting set of studies with relevance to a broad range of disciplines. Of particular interest is the examination of the impacts of maternal behavior and social environment on epigenetic marks and corticosteroid levels in a complex social and ecological environment. As so much of both the epigenetic and stress literature relies on either rodents in highly deprived artificial environments or human studies which have to cope with high degrees of unobserved variability and reliance on self-report, the study should help resolve some long standing questions with regard to development and stress. I'm particularly struck by the fact that maternal behavior has such a limited effect on adult corticosteroid levels, while the richness of social contacts outside of the den appears to be more powerful. I think this may begin to explain why so many early life interventions in humans fail to produce substantial effects across the lifespan, and point the way to new developmental windows in which to intervene. The paper was well written and should be published with a commentary calling attention to it. I have only a few substantive concerns.

Response: Thank you for thoughtfully reviewing this manuscript. We are grateful that the reviewer is excited about this work, and in particular, appreciates the value of the wild study population.

2. **Comment:** P-7. Description of the LUMA assay- states: "The majority of CpG sites assessed...occur in gene bodies ... as well as in non-coding regions of the genome..." As this description basically covers the entire genome, I'm not sure how meaningful the argument is. I think it might be better to simply cite some references that associate global DNA methylation with either health, epigenetic age or some equivalent variable.

Response: This is a salient point and germane to our discussion of the results. Reviewer 1 has also noted this issue, so we refer to our previous responses to Reviewer 1's comments #2, #7, and #10. More specifically, in Rev. 1, comment #2 we discuss associations between global DNA methylation and various health outcomes and provide additional citations in text. In Rev. 1 comments #7 and #10, we present new analyses that annotate the 'CCGG' motif in the recently published draft spotted hyena genome. These new analyses and associated figures provide a much more detailed description of what the LUMA assay measures in this species.

3. **Comment:** Following the thought above. Would it be possible to examine epigenetic age in these samples? If so I think it would substantially enhance the paper.

Response: In the present study, our goal was to assess potential epigenetic markers of early social experience via DNA methylation. While epigenetic age is certainly interesting and may contribute to the relationships assessed herein, it is a distinct idea that we hope to assess in a future study with a larger sample of mERRBS data.

4. **Comment:** For the EWAS analysis (and more generally), why was the domestic cat used when a draft genome of the both the striped and spotted hyenas exist (Yang et al 2020)? Even if these draft assemblies are too spotty to be useful in all cases, it should be noted they do exist and explained why they were not used.

Response: We have clarified in text that our DNA methylation calling as part of our EWAS was done using the BGI hyena genome, which was unpublished at the time of initial submission. We have now added the appropriate citation.

Methods:

Pg 8, Lines 137-139:

After DNA sequencing, we ran a standard bioinformatic pipeline to clean, align, and call DNA methylation reads using the draft spotted hyena genome (Yang et al., 2020).

5. **Comment:** Similarly, it would increase my confidence in the data if the approach of using the cat genome was validated using another well-annotated carnivore such as the domestic dog.

Response: Thank you for pointing out our missing citation for the draft hyena genome. As mentioned above, we used the draft hyena genome for DNA methylation calling and we used cat and human genomes to determine nearest genes in our post EWAS analyses.

Comment: With regard to the glucocorticoid receptor data, is it possible the site examined was a pseudogene? If not, it might be useful to give a bit more context in terms of hyena endocrinology, which is relatively unique, and might explain why this species appears to have no plasticity at this gene, which seems to have plastic expression/epigenetic regulation in most other mammalian species thus far examined. Yang C, Li F, Xiong Z, Koepfli KP, Ryder O, et al. 2020. A draft genome assembly of spotted hyena, *Crocuta crocuta*. *Sci Data* 7: 126

Response: We Sanger sequenced part of the putative *GR* gene in hyenas. Similar to the homologous regions in humans and rodents, the hyena amplicon contains the start codon sequence, ATG, which is sometimes missing in pseudo-genes. Furthermore, our pyrosequencing assay included 6 CpGs that were shown to be differentially methylated in both humans and rodents.

Regarding the lack variation/plasticity in DNA methylation in hyenas: we suspect that this may partially reflect the fact that we used leukocytes instead of brain tissue, the latter of which seems to be more variable in *GR* DNA methylation. To address this, we exercise caution in interpreting our null results and suggest future studies consider a more complete sequencing and testing of differential CpG methylation across the entire *GR* gene.

Discussion:

Pg 38, Lines 652-655:

Future studies may better replicate results from laboratory rodents by using brain tissue, wherein expression of the *GR* gene is more tightly coupled with HPA regulation or by exploring DNA methylation of additional CpGs with more extensive coverage of the hyena *GR* gene.

Reviewer #3

1. **Comment:** Laubach et al.'s interesting manuscript entitled Associations of early social experience with offspring DNA methylation and later life stress phenotype represents an important investigation of the role of social interactions/connectedness on subsequent genetic (DNA methylation) and stress responses (fecal-based glucocorticoid metabolites). The use of a novel model species ---the spotted hyena---in a naturalistic setting brings much needed authenticity to previous investigations in this research area that are based on laboratory bred animals raised in restricted laboratory environments. Further, the importance of social interactions in natural hyena behavior and the relevance of their social hierarchy on survival-related resources make this species ideal for the proposed research in the current study. It's not surprising that the data depart slightly from earlier findings in related studies focusing on maternal behavior and subsequent hypothalamic-pituitary-adrenal (HPA) activity in developing offspring. For example, it was reported in Laubach's manuscript that observations of maternal care early in hyena offspring development seemed to be less impactful than reported in rats [e.g., by Michael Meaney's team's work on low-licking and high-licking maternal rats (e.g., see Caldji et al., 1998, Maternal care during infancy regulates the development of neural systems mediating the expression of fearfulness in the rat. Proceedings of the National Academy of Sciences of the United States of America, 95(9), 5335–5340)]. However, by focusing further in the developmental phase (i.e., Den Independence phase), social interactions were indeed found to be associated with fecal metabolites in a direction indicating that increased social interactions mitigated the glucocorticoid response. Still, Laubach et al.'s findings suggest that the maternal-offspring interactions may be more complex than previously observed in the rodent studies. For example, it is possible that more vulnerable offspring may be viewed to elicit more maternal responses than healthier animals when intra- and inter-litter differences are considered. The limitations of collecting fecal samples at various times throughout the day were addressed by treating the time of collection as a covariate in the statistical analyses.

Response: We thank the reviewer for their time and effort spent vetting this work. We are pleased that they find this analysis to yield interesting findings.

2. **Comment:** Looking forward, it would be interesting to also include DHEA assays when processing fecal samples, as considering DHEA/CORT ratios provide another index of adaptive stress responsivity since DHEA has been associated with emotional resilience, among other functions (e.g., see: Sripada RK, Marx CE, King AP, et al. DHEA enhances emotion regulation neurocircuits and modulates memory for emotional stimuli. Neuropsychopharmacology. 2013;38(9):1798-1807). The fact that the samples weren't collected after a stress challenge limits the information about responsivity but this is addressed in the manuscript. Looking back in the data, there may have been isolated times after a natural stressor that may be interesting to try to tease apart from the pooled baseline data.

Response: Thank you for this suggestion. We agree that there are a number of additional metrics of stress responsivity that would provide valuable insight and potentially help disentangle the adaptive function of developmentally plastic stress physiology. We noted that we were not able to measure change in stress hormone levels in response to a standardized challenge as a limitation, and we suggest that future studies should consider quantifying this difference in response to the darting event as has been done by Dr. Robert Sapolsky in baboons. Quantifying DHEA and modeling the ratio of CORT/DHEA is another good idea. We have added this suggestion in the discussion of future directions.

Discussion:

Pg 39, Lines 677-680:

However, a more nuanced measure of stress, like reaction and recovery to a standardized stressor or the ratio of cortisol to dehydroepiandrosterone (CORT/DHEA), may provide a more direct assessment of acute HPA function (Kamin & Kertes, 2017; Romero, Dickens, & Cyr, 2009) over time.

3. **Comment:** Focusing on the findings that higher social connectedness during the maternal and later developmental phase affects global DNA methylation corroborate previous work, and enhances the robustness of these earlier findings with ecological relevant data in a unique species model.

Response: We agree and thank the reviewer for this comment. We have slightly modified the text to clarify this point.

Discussion:

Pg 33, Lines 547-551:

Greater maternal care (specifically, close proximity received during the first year of life) and higher social connectivity (namely, degree during the DI life stage) were both associated with higher global DNA methylation later in life. These results corroborate those of previous studies that report associations of early social experiences with differential patterns of DNA methylation in rodents, Rhesus macaques, and humans (Anier et al., 2014; Provencal et al., 2012; Unternaehrer et al., 2015).

4. **Comment:** Concerning the scoring of maternal behavior, it wasn't clear if proximity of the infant to the mother was viewed independently from the nursing and grooming behavioral categories. Was proximity scored when the animals were in proximity but NOT engaging in those behaviors? Was social play considered in the early behavioral observations?

Response: Thank you for the opportunity to clarify. Close proximity includes time spent nursing or grooming. We have updated the text in the supplemental information to reflect this detail and discussed this as a potential reason for the strong associations of close proximity with global DNA methylation. Social play was not considered in these behaviors.

Discussion:

Pg 34, Lines 560-565:

One potential explanation for the robust effect of close proximity but not the other metrics of maternal care (nursing and grooming) on global DNA methylation may reflect the fact that close proximity includes the time mother hyenas spent nursing and grooming. Thus, close proximity is a composite metric of maternal investment in offspring that may capture more variation in offspring global DNA methylation.

Supporting material:

Pg 1 (SI), Lines 9-15:

We focused on three specific maternal care behaviors: time spent in close proximity, nursing, and grooming. We chose these specific metrics of maternal care because comparable behaviors in primates and rodents have been shown to influence offspring behavior and physiology, albeit in captive settings. Additionally, these behaviors capture the effect of nutrition as well as physical contact on offspring development. *Nota bene*, maternal care behaviors were not mutually exclusive in their occurrence; for example, a portion of time spent in close proximity includes time spent nursing or grooming.

5. **Comment:** The sophistication of the statistical approaches in this manuscript (e.g., linear mixed model) is impressive and appears to consider the most likely causal paths with the social connectedness-stress response scenario. With the data being drawn from a pool of observations/samples extending from 1988-2016, is it necessary to describe other studies published from these data? I see that one study apparently published from this data set is mentioned in the current manuscript (i.e., Laubach et al., 2019. *Mol. Ecol.*), but doesn't explicitly convey that the findings were in the same group of animals/data.

Response: Thank you. The reviewer is correct that this analysis is based on a subsample of animals who are part of a long-term study of wild hyenas. Some of the individuals in the present analysis were also in previous analyses and are published in manuscripts, including Laubach *et al.* 2019. *Mol. Ecol.* 11912: 1413-1427. We cite these previous publications in text.

Methods:

Pg 7, Lines 104-107:

For each hyena, we constructed association networks during the CD and DI periods (two life stages during which social interactions were previously identified as key determinants of fitness²⁹) as previously described for this population ²⁶.

Pg 7, Lines 119-121:

Descriptions of the LUMA assay, laboratory procedures, and data cleaning protocol are available in (Laubach, Faulk, et al., 2019), from which the current global DNA methylation samples were drawn.

6. **Comments:** To succinctly address the prescribed questions highlighted for reviewers, the findings described in the current manuscript are indeed noteworthy as they provide a naturalistic, ethologically-relevant approach to genetic and endocrine markers of emotional resilience and stress. This approach qualifies specific aspects of past findings with rodent models. The findings support the claims in the study and, although the methodology has limitations in areas (e.g., not being able to control time of fecal sample collections to control for circadian effects), the authors cleverly compensated for these variations with statistical techniques. The methods are clearly described in the manuscript and supplemental materials.

Response: Thank you, again.

Reviewer #4:

1. **Comment:** In the manuscript entitled “Associations of early social experience with offspring DNA methylation and later life stress phenotype” the authors chose wild spotted hyenas (*Crocuta crocuta*) as model species aiming to decipher the impact of early life experiences on faecal glucocorticoid metabolite levels and DNA methylation. The authors also test the hypothesis that DNA methylation may act as physiological mediator among early life experience and stress later in life. The study was inspired by former studies in laboratory model species under controlled experimental set-ups. Early experiences were measured by observation of maternal care which were specified as grooming, nursing and close proximity to the offspring’s mother during two chosen time windows in early development “communal den (CD) phase” and the subsequent “den-independent (DI) phase”.

Response: Thank you for the detailed and timely review of our manuscript. These comments have been insightful, and no doubt have improved the manuscript.

2. **Comment:** Unfortunately, it remains unclear why these specific windows were chosen, as well as their lengths in months.

Response: The time-windows of communal den (CD) and den independent (DI) were chosen to reflect developmental stages that correspond with key life history events in spotted hyenas and are based on previous studies in our population that linked early life social experiences to long-term health and fitness. More specifically, we defined the CD period based on data from daily observations of individual hyenas and their activity patterns, which were confined to the communal den. We also recorded when young hyenas began forays away from the communal den out into their clan’s territory, which we noted as the start of the DI period. The duration of the DI period was matched to the duration of the CD period in order to balance our sampling design, as previously done in this study population (Turner *et al.* 2018. *Behav.Eco. Soc.* 72). Furthermore, social networks metrics assessed within the CD and DI phases using this method are associated with fitness in this population, supporting the biological relevance of these metrics (Turner *et al.* 2021. *J. Anim. Eco.* 90: 183-196). We have updated the main text and the text in the Supporting Material to clarify our justification of the time-windows used in our analyses.

Methods:

Pg 7, Lines 104-107:

For each hyena, we constructed association networks during the CD and DI periods (two life stages during which social interactions were previously identified as key determinants of fitness²⁹) as previously described for this population ²⁶.

Supporting material:

Pg 1 (SI), Lines 4-7:

Our maternal care data included 1533 FAS totaling approximately 779 hours of observations of 258 mother-infant pairs when offspring were ≤ 1 year old, which is the approximate age at weaning in this sample (11.59 months) and among spotted hyenas more generally (Holekamp & Smale, 1998).

3. **Comment:** The biological meaning of splitting “maternal care” into the three variables: grooming/licking, nursing and close proximity needs further explanation.

Response: We have added additional explanations in the supporting material to address this comment.

Supporting material:

Pg 1 (SI), Lines 9-15:

We focused on three specific maternal care behaviors: time spent in close proximity, nursing, and grooming. We chose these specific metrics of maternal care because comparable behaviors in primates and rodents have been shown to influence offspring behavior and physiology, albeit in captive settings. Additionally, these behaviors capture the effect of nutrition as well as physical contact on offspring development. *Nota bene*, maternal care behaviors were not mutually exclusive in their occurrence; for example, a portion of time spent in close proximity includes time spent nursing or grooming.

- Comment:** Figure 6 shows that only close proximity of offspring to the mother has an effect on global methylation, but neither grooming nor nursing. The biological meaning of this but not the others remain unexplained in the discussion, where the authors refer to this result as maternal care, instead of to close proximity. “We also found that more maternal care in the first year of life, which roughly corresponds with the CD period of development,”. The same strategy of gathering the results which were before specified holds true for measures of the CD period, stated in the second part of this sentence “...and greater social connectedness during the DI phase were each associated with higher global DNA methylation, a presumed indicator of genomic stability and overall health.”, while figure 7 shows only a slight effect for the “degree” of interactions on the global methylation ratio, but not the others. Why were these variables separated into more specific variables, but explained as one? What is the biological relevance of these specifications?

Response: Thank you for bringing this point to our attention. We have revised our text in the discussion so that our explanations reflect findings with respect to specific early life social experiences, and to address the reason why we may have observed an association with close proximity but not the other maternal care variables.

Discussion:

Pg 33, Lines 547-551:

Greater maternal care (specifically, close proximity, received during the first year of life) and social connectivity (namely degree during the DI life stage) were both associated with higher global DNA methylation later in life, enhancing the robustness of previous findings that suggest early social experiences influence patterns of DNA methylation (Anier et al., 2014; Provencal et al., 2012; Unternaehrer et al., 2015).

Pg 34, Lines 560-565:

One potential explanation for the robust effect of close proximity but not the other metrics of maternal care (nursing and grooming) on global DNA methylation may reflect the fact that close proximity includes the time mother hyenas spent nursing and grooming. Thus, close proximity is a composite metric of maternal investment in offspring that may capture more variation in offspring global DNA methylation.

- Comment:** Furthermore, the specific maternal care observatory data were measured in 1 to 29 repeated observations of mother-offspring interactions. These variable estimated values were subsequently reduced to a single value (by calculating the distance to the mean of each measurement), erasing its variation, and thus reducing the complexity of the data. This approach has been shown to be problematic as this strategy leads to a decrease in robustness of the model these values are used in (<https://academic.oup.com/beheco/article/28/4/948/3059669>).

Response: Indeed, the paper by (Houslay & Wilson. 2017. *Behav. Eco.* 28: 948-952) posits that use of BLUPs as a data reduction technique for the dependent variable of interest may artificially shrink confidence intervals to confer a larger degree of precision than use of use of mixed effects models with random effects to account for correlations between repeated measurements (or clusters of data points). However, a key distinction is that we use BLUPs to summarize repeated measures of the explanatory variables (maternal care behaviors) - not the dependent variable of interest. This data reduction step is necessary for modeling repeated measures data as the independent variable, does not confer any sort of shrinkage effect on the estimate of interest, and has been used in other similar settings (e.g., repeated measures of four LINE-1 DNA methylation sites in an analysis of LINE-1 DNA methylation as a predictor of subsequent growth: Perng *et al.* 2013. *PLoS One.* 8: e62587). More specifically, we used this data reduction technique because of the varying number of times (1 to 29 times) that each mother-offspring pair was observed, and because it would be very challenging to assess and interpret associations of behaviors at each individual time-point in relation to the dependent variables of interest.

6. **Comment:** Also it is unclear why certain covariates were used in respective models e.g. one includes social rank, but not the others. Please explain how they were chosen.

Response: We selected covariates for multivariable models based on prior knowledge on determinants of stress phenotype (including use of the directed acyclic graphs to visualize the hypothesized temporal and causal associations among variables of interest) and bivariate associations (**Supporting Tables 2-5**). This approach is widely-used by epidemiologists to efficiently and methodically make causal inference from observational data, as discussed in a recent review that our group led (Laubach *et al.* 2021. *Proc. Royal Soc. B.* 288: 2020815). We now mention this in the Methods section (**Methods: Page 10, Lines 167-168**)

7. **Comment:** The offsprings' sex seems critical, because females are philopatric while males disperse. As a result it may lead to differences in early behaviour. Additionally, puberty in males occurs earlier (after 18-24 months) than in female spotted hyenas (after 24-30 months), which may also lead to differences in early behaviour, and has been shown to change the overall DNA methylation in human. Please provide the number of males and females in the manuscript.

Response: Done (**Page 5, Lines 75-77; and in table and figure legends**)

8. **Comment:** A comparison of a wild mammal to non-model species seems like the attempt to simplify a highly complex system within an uncontrolled experimental set-up. This simplification is critical. The authors are partly aware of this as they state within the discussion "..., in contrast to controlled rodent experiments, wild hyenas are subject to a multitude of stressors over development, which may hamper our ability to isolate the specific effect of maternal care on stress phenotype. Second, compared to maternal separation common in many experimental studies, natural variation in maternal care is far more subtle and may require a more sensitive measure of physiological stress than average fGCMs, which is a summary baseline stress indicator 51 subject to unmeasured environmental factors and the animal's condition when the sample is collected 52."

Response: Thanks for the suggestion. We have added the following text:

Discussion:

Pg 33, Lines 540-543:

Considering not only the differences in how early-life experience and stress outcomes are measured, but also differences between controlled laboratory settings vs. studies of wild animals, we urge caution when comparing our results to previous studies involving captive primates and rodents.

9. **Comment:** The measurement of faecal glucocorticoid metabolite is a valid method. But stress levels are prone to outer environmental changes e.g. predatory attacks and human interactions increase stress levels. Accounting for these events is often difficult due to limited time windows of observation. “Behavioral data were collected daily between 0530 – 0900 h and 1700 – 2000 h.” Therefore replicates of fGCMs for each animal are important as the period and day of sampling is critical. Please state if they were applied.

Response: Unfortunately, we do not know if hyenas experienced highly stressful events (e.g. encounter’s with lions or Masai pastoralists) prior to defecating due to limited time windows of observation, and due to hyenas’ fission-fusion social structure, which means that observers are never observing all the hyenas at once. However, we measured fGCMs, which represent stress levels over the course of hours, from multiple samples in order to calculate an average level of the stress hormone that minimizes the effect of any particular acute experience, such as predatory attacks or out-of-the-ordinary interactions with humans. Accordingly, we feel that our lack of data on such events does not pose a major issue.

Regarding variation in fGCMs that result from natural cycles in steroid hormones over the course of the day, we include time of day at which the fecal sample was collected in our models. This is noted in the results and the footnote of **Table 1**.

10. **Comment:** Global methylation analysis using LUMA only provides a methylation ratio over the entire genome. It does not provide the data for functional interpretation. The authors state “The majority of CpG sites assessed via LUMA occur in gene bodies, where they may function in transcription regulation and alternative splicing 27, as well as in non-coding regions of the genome, where they may repress repetitive elements 28 and enhance chromosome stability 29. Therefore, we assumed that lower than average %CCGG methylation is likely disadvantageous to health.” This assumption is very farfetched, because LUMA does not provide the details to distinguish between functional important regions and mostly excludes promoters, which are thought to be the main regulatory regions. Therefore methylation changes are difficult to interpret and should be handled with caution. Using LUMA is cost-efficient and does not need a reference genome, which would be two valid reasons to use it. However the usage of three different approaches with different samples numbers: global DNA methylation (n = 186), genome-wide DNA methylation (n = 29), and candidate gene DNA methylation (n = 96) without explanation, is confusing. Please clarify why you chose this strategy.

Response: Thank you for raising these important points. First regarding the issue of interpreting our global DNA methylation results we kindly refer the reviewer to our responses to Reviewer 1, Comments #2 and 3

Regarding the use of multiple assays and differing sample sizes for the assays: each assay presents a tradeoff, some of which are logistical, including the lack of an available reference genome, and cost efficiency as noted by the reviewer. Other reasons for using multiple assays are biological and include interpretation of results that reflect broad scale genomic stability versus functional pathways vs potential candidate genes and their influence on phenotype. We included additional analyses and

results that more explicitly highlight what each assay measures and provide an overview of the genomic context that is targeted by each assay. The revised text and new results and figures are in our response to Reviewer 1, Comment 7.

11. **Comment:** In order to assess the biological meaning of the established data, please clarify the numbers of animals, mothers, sex of offspring, and siblings, and the pairwise interactions observed. The number of samples collected for CD and DI time windows, stating why these were chosen, the sex, the cofounding factors that can lead to an increase in stress, the robustness of the metabolites measured. Last but not least, providing the data and code used is essential to understand and repeat the analysis.

Response: Regarding *“the numbers of animals, mothers, sex of offspring, and siblings, and the pairwise interactions observed”*: we report the requested information for each data set in the main text and provide additional information regarding sample sizes and the overlap between different data sets (i.e. the numbers of animals included in different models) in the supplemental information and table/figure legends.

Regarding *“the number of samples collected for CD and DI time windows, stating why these were chosen, the sex, the cofounding factors that can lead to an increase in stress, the robustness of the metabolites measured”*: We have previously addressed this issue in our response to Reviewer 4 Comment #2.

Regarding *“Last but not least, providing the data and code used is essential to understand and repeat the analysis”*: We will submit our code and related data sets to a public GitHub repository such that anyone can reproduce these analyses or implement a similar approach using their own data. We agree that making code available is important for reproducibility, and we have done this with previously published projects that use these data (c.f. the code repository for the analyses in Laubach *et al.* 2019. *Molec. Eco.* 28: 3799-3812, located here: https://github.com/laubach/hy_luma). We will also follow additional guidelines regarding data archiving as set forth by the journal.

REVIEWER COMMENTS

Reviewer #1 (Remarks to the Author):

Overall, the authors succeed in addressing all but two of my concerns. I feel their revisions (as well as the strong original manuscript) will make an important contribution to our understanding of stress biology and early life adversity.

The following two points have not been adequately addressed:

- I'm not convinced by the authors' response to concerns regarding cell type composition. The problem with changes in cell type composition isn't that they produce local effects at individual genes, but rather systemic biases. They therefore obscure true signal of specific genes or biological pathways, with signatures of cell-type differences. New citations to Schisterman and colleagues represent a minority and outdated opinion in the field.

If the authors still have access to the primary samples, I would suggest they use blood smears to estimate cell type composition for at least the EWAS analysis. If these samples are exhausted, I would remove the references to Schisterman et al. and leave the revised text as "Fourth, we used archived DNA from whole blood that lacked information on cell type composition, and therefore we were not able to control for cell type heterogeneity as a source of variation in DNA methylation. However, cell type composition may be influenced by factors like social stress (Engler et al., 2004), and therefore could be on the causal pathway between the explanatory variables of interest and DNA methylation."

- Second, I repeat my suggestion that the authors consider systematic biases in the effect sizes across genomic contexts to add insight into the global methylation patterns. This was partially addressed for significantly differentiated sites [lines 453-455], but not for all sites which may reveal subtle, systematic effects that underlie the global methylation effects.

In response to my original comment, the authors stated "the present analysis is not an ideal setting in which to formally compare global vs genome-wide results for a few reasons. First, the overlap in the global and genome-wide DNA methylation samples includes only 12 animals, therefore a direct comparison between data sets is limited. Second, the purpose of the EWAS analysis was exploratory, and to identify potential functional pathways that can be assessed more rigorously in future analyses with larger sample sizes."

The first point seems to be an advantage rather than a disadvantage. If the patterns observed in the larger, global DNA methylation dataset are robust we would expect them to be consistent with the genome-wide methylation dataset, even for completely independent datasets. Second, the exploratory nature of these EWAS results does not prevent them from being used for this purpose. In fact, directional biases in effect size and analyses by genomic context are in many ways less problematic than site-specific analyses using a small sample size. Finally, while I appreciate concerns regarding manuscript length, I think such an analysis could be a concise addition to the main text, such as: "Furthermore, our

EWAS results reveal a bias in reduced methylation levels, consistent with our global analysis of DNA methylation. This effect is largely driven by reduced methylation in XXX genomic contexts, where XX contexts have no directional bias”.

Reviewer #2 (Remarks to the Author):

I thank the authors for their thoughtful revisions. I look forward to seeing the paper in print.

All the Best,
Richard Hunter, Ph.D.

Reviewer #3 (Remarks to the Author):

As conveyed in my initial review of this manuscript, I feel that it represents a valuable body of work that will be extremely informative as we reconcile laboratory and field studies targeting genetic and stress response perspectives. After reviewing the authors' thoughtful and thorough responses to the reviewer comments, I am satisfied that all points were appropriately considered and included in the revised manuscript.

Reviewer #4 (Remarks to the Author):

Reviewers comment to Response:

Thank you for the authors' additional work and replies to the reviewers' comments.

After re-reading the publication of Houslay & Wilson (2017. *Behav. Eco.* 28: 948-952) on the misuse of BLUPs, I remain doubtful that the method used by the authors is appropriate, as it hides uncertainty. The authors' response does not clarify the statistical or logical reason justifying that the problem highlighted by Houslay & Wilson doesn't apply, when BLUPs are used in subsequent analysis as explanatory variable. In the referred article I could not find any instance implying such a statement. Rather I read: “[...], it has become common practice to extract predictions of individual random effects from fitted mixed models and to use these in subsequent analyses, such as correlation tests or linear regression models (Table 1). Problems arise from this approach because individual point estimates from random effects in mixed models (sometimes known as conditional modes, or best linear unbiased predictors, BLUPs) are predicted with large amounts of error. Their use in secondary analyses can therefore lead to highly anticonservative tests of biological hypotheses, because the error inherent in their prediction is excluded from these further tests (Hadfield et al. 2010).” Houslay & Wilson further state “We stress that BLUP is an incredibly useful technique that should not be dismissed in any way as

inherently “bad” (Robinson 1991). Indeed, it is entirely appropriate to use individual-level predictions to say something about individuals (or genotypes, or specific levels of some other random effect).”, and “Nonetheless, use of these “stats on stats” approaches that are known to be inappropriate for hypothesis testing (see Brommer 2013b for further discussion) continues unabated.” Using stats on stats should be avoided, as data might be highly fragile to changes e.g. after an increase in sample number. Thank you for adding the sample numbers, and including numbers of males and females. “Using blood samples from immobilized hyenas, we 74 constructed three primary datasets for analyses: global DNA methylation (n = 186 total; n = 99 75 females and n = 87 males), genome-wide DNA methylation (n = 29 total; n =29 females and n = 76 0 males), and candidate gene DNA methylation (n = 78 total; n = 43 females and n = 35 males).” With respect to these numbers, the experimental set-up seems not fully thought through beforehand and remains confusing. For example, for the analysis of DNA methylation three methods were applied using different individuals, amounts and different sex composition. Sex is, as pointed out before, an essential determiner within the hyena system, which as such has an effect on reproduction, behaviour and thus on DNA methylation. Furthermore, the usage of whole blood samples is tricky, blood-cell composition quickly changes e.g. in case of an infection numbers of leucocytes are increasing. The cell composition change is accompanied by a change in methylation. The authors cannot control for this – as stated in their response to reviewer 1.

Despite the authors` best effort, the results look fragile and dependent on the method used. I suggest applying a measure of sensitivity or even more holistic approach, such as a multiverse analysis proposed by Gelman (<https://doi.org/10.1177/1745691616658637>), to increase the study's significance.

While the idea and species` system is exciting, at this stage I am not convinced the results provide deeper knowledge into the species system.

Point-by-point response to Reviewer comments

Reviewer #1

1. **Comment:** Overall, the authors succeed in addressing all but two of my concerns. I feel their revisions (as well as the strong original manuscript) will make an important contribution to our understanding of stress biology and early life adversity.

Response: We are pleased that we have addressed the majority of this reviewer's concerns and look forward to making a formal contribution to the literature.

2. **Comment:** I'm not convinced by the authors' response to concerns regarding cell type composition. The problem with changes in cell type composition isn't that they produce local effects at individual genes, but rather systemic biases. They therefore obscure true signal of specific genes or biological pathways, with signatures of cell-type differences. New citations to Schisterman and colleagues represent a minority and outdated opinion in the field. If the authors still have access to the primary samples, I would suggest they use blood smears to estimate cell type composition for at least the EWAS analysis. If these samples are exhausted, I would remove the references to Schisterman et al. and leave the revised text as "Fourth, we used archived DNA from whole blood that lacked information on cell type composition, and therefore we were not able to control for cell type heterogeneity as a source of variation in DNA methylation. However, cell type composition may be influenced by factors like social stress (Engler et al., 2004), and therefore might be on the causal pathway between the explanatory variables of interest and DNA methylation."

Response: Thank you. We have included the suggested language in the discussion of limitations.

Pg 39-40, Lines 761-765: Fourth, we used archived DNA from whole blood that lacked information on cell type composition, and therefore we were not able to control for cell type heterogeneity as a source of variation in DNA methylation. However, cell type composition may be influenced by factors like social stress¹⁰², and therefore might be on the causal pathway between the explanatory variables of interest and DNA methylation.

3. **Comment:** Second, I repeat my suggestion that the authors consider systematic biases in the effect sizes across genomic contexts to add insight into the global methylation patterns. This was partially addressed for significantly differentiated sites [lines 453-455], but not for all sites which may reveal subtle, systematic effects that underlie the global methylation effects. In response to my original comment, the authors stated "***The present analysis is not an ideal setting in which to formally compare global vs genome-wide results for a few reasons. First, the overlap in the global and genome-wide DNA methylation samples includes only 12 animals, therefore a direct comparison between data sets is limited. Second, the purpose of the EWAS analysis was exploratory, and to identify potential functional pathways that can be assessed more rigorously in future analyses with larger sample sizes.***" The first point seems to be an advantage rather than a disadvantage. If the patterns observed in the larger, global DNA methylation dataset are robust we would expect them to be consistent with the genome-wide methylation dataset, even for completely independent datasets. Second, the exploratory nature of these EWAS results does not prevent them from being used for this purpose. In fact, directional biases in effect size and analyses by genomic context are in

many ways less problematic than site-specific analyses using a small sample size. Finally, while I appreciate concerns regarding manuscript length, I think such an analysis could be a concise addition to the main text, such as: “Furthermore, our EWAS results reveal a bias in reduced methylation levels, consistent with our global analysis of DNA methylation. This effect is largely driven by reduced methylation in XXX genomic contexts, where XX contexts have no directional bias”.

Response: Thanks to the reviewer for taking the time to clarify what they would like to see here. We now show patterns of positive and negative associations between DNA methylation and fecal corticosterone, comparing distributions of beta estimates within genomic regions vs across the whole genome (**SI Figure 11**; pasted below). We have also added text in the Methods & Results sections describing this figure.

Methods:

Pg 18, Lines 350-355: Finally, among the 25 individuals for which we assayed mERRBS, we annotated CpG sites from the EWAS in genomic regions of the draft hyena genome as described in the Supporting Material. We then assessed patterns of positive and negative associations of DNA methylation with fGCMs and compared the distribution of *Beta* estimates within genomic regions vs. across the entire genome to assess for bias in the direction of associations from the EWAS.

Results:

Pg 26, Lines 499-503: While we did not observe systemic bias in enrichment of positive or negative associations from our EWAS, within particular genomic regions, including CpG islands, promoters and introns there appeared to be fewer *Beta* estimates near zero than compared to all CpG sites across the whole genome (**Supporting Figure 11**).

Supporting Figure 11. Density plots of Beta values from the stress hormone EWAS after bias and inflation correction. All Beta values (white histogram) are overlaid with Beta values from specific annotation regions (gray filled histogram). **a)** Beta values from stress hormone EWAS based on

the CpG density annotation. **b)** Beta values from stress hormone EWAS based on genic annotation.

In addition, we agree that in a scenario where maternal care is positively associated with global DNA methylation, then it would be interesting to examine systematic associations/biases within genomic contexts identified by the EWAS. In doing this, we expect that a comparison of counts of loci with positive vs. negative associations between maternal care and (site-specific) DNA methylation would reveal genomic contexts driving the association captured by the global DNA methylation assay. This approach would be informative if we were modeling the same predictor of global DNA methylation as in the EWAS analysis. However, this is not the case for the present study. In this analysis, we examined associations of multiple early-life social experiences and global DNA methylation as part of a broader analysis that sought to test global DNA methylation as a possible mediating mechanism linking early-life social experiences to stress physiology. Because we did not find any associations of global DNA methylation and stress physiology (fecal glucocorticoids), we did not continue the assessment of mediation. Instead, we implemented an outcome-based EWAS (i.e., an EWAS that identified hits with respect to stress physiology) to identify genomic sites/regions associated with fecal glucocorticoids, and then used a meet-in-the-middle approach to identify those that are also associated with maternal care as an indicator of the early-life social experience (NB: we selected maternal care based on prior knowledge and findings in this study population). Since we did not conduct an EWAS in which early-life social experience is the explanatory variable, a direct comparison will not be informative.

Reviewer #4

1. **Comment:** Thank you for the authors' additional work and replies to the reviewers' comments. After re-reading the publication of Houslay & Wilson (2017. *Behav. Eco.* 28: 948-952) on the misuse of BLUPs, I remain doubtful that the method used by the authors is appropriate, as it hides uncertainty. The authors' response does not clarify the statistical or logical reason justifying that the problem highlighted by Houslay & Wilson doesn't apply, when BLUPs are used in subsequent analysis as explanatory variable. In the referred article I could not find any instance implying such a statement. Rather I read: "[...], it has become common practice to extract predictions of individual random effects from fitted mixed models and to use these in subsequent analyses, such as correlation tests or linear regression models (Table 1). Problems arise from this approach because individual point estimates from random effects in mixed models (sometimes known as conditional modes, or best linear unbiased predictors, BLUPs) are predicted with large amounts of error. Their use in secondary analyses can therefore lead to highly anticonservative tests of biological hypotheses, because the error inherent in their prediction is excluded from these further tests (Hadfield et al. 2010)." Houslay & Wilson further state "We stress that BLUP is an incredibly useful technique that should not be dismissed in any way as inherently "bad" (Robinson 1991). Indeed, it is entirely appropriate to use individual-level predictions to say something about individuals (or genotypes, or specific levels of some other random effect).", and "Nonetheless, use of these "stats on stats" approaches that are known to be inappropriate for hypothesis testing (see Brommer 2013b for further discussion) continues unabated." Using stats on stats should be avoided, as data might be highly fragile to changes e.g. after an increase in sample number.

Response: We thank the reviewer for providing additional comments and suggestions on this manuscript. We agree that use of BLUPs to simplify repeated measures as the dependent variable is problematic due the reasons highlighted by the reviewer. However, these issues and those raised by Houslay & Wilson do not apply to the present analysis where BLUPs are used to represent inter-individual variation in repeated measures of the **independent (aka explanatory variable)**. This is known as the "two step approach," a well-accepted method to reduce dimensionality of repeated measures of the independent variable of interest (c.f., Chen et al. *Environ Health.* 2015;14(9) doi: 10.1186/1476-069X-14-9; Villarroel et al. *Stat Med* 2009; 28(20:2552-2565). Specifically, prior to analyzing maternal care (assessed via multiple repeated behavioral observations) as a determinant of offspring DNA methylation or fecal glucocorticoid levels, we first consolidated the maternal care behaviors into a single estimate for each mother-offspring dyad via BLUPs extracted from a mixed-effects regression. These BLUPs represent the average deviation of maternal care for each mother-offspring dyad from the population average, effectively capturing inter-individual variation in maternal care. We then entered these BLUPs into a subsequent model as the explanatory variable of interest, in relation to offspring DNA methylation or fecal glucocorticoid concentrations. We also note that this is likely the most informative and biologically relevant option for analyzing explanatory variables that are assessed repeatedly over time. Alternate approaches include: (1) calculating average values across all repeated measurements of maternal care, which would result in loss of variability and information; (2) selecting a single representative measure, which entails arbitrary decisions regarding which measure is best (for which we do not have a set of consensus criteria); or (3) creating a latent variable via ordination techniques which are feasible but hamper interpretability. Given the limitations of these alternative approaches, we strongly feel that our use

of BLUPs to summarize maternal care behaviors permits the most straightforward biological interpretation, maximally utilizes the data and available information, and is statistically conservative.

2. **Comment:** Thank you for adding the sample numbers, and including numbers of males and females. “Using blood samples from immobilized hyenas, we 74 constructed three primary datasets for analyses: global DNA methylation (n = 186 total; n = 99 75 females and n = 87 males), genome-wide DNA methylation (n = 29 total; n =29 females and n = 76 0 males), and candidate gene DNA methylation (n = 78 total; n = 43 females and n = 35 males).” With respect to these numbers, the experimental set-up seems not fully thought through beforehand and remains confusing. For example, for the analysis of DNA methylation three methods were applied using different individuals, amounts and different sex composition. Sex is, as pointed out before, an essential determiner within the hyena system, which as such has an effect on reproduction, behaviour and thus on DNA methylation.

Response: We agree that differences and overlaps in sample sizes is not ideal. This is due, in large part, to the fact that this analysis is in fact not an experiment but rather, an analysis of opportunistically collected observational data. As noted by the reviewer, we measure DNA methylation using three separate assays that provide unique information including global DNA methylation, CpG site specific DNA methylation across the genome, and a candidate gene approach. Samples for each assay were based on a combination of opportunistic dartings for which feasibility was determined by constraints of field work and matched with existing behavioral and stress physiology measures.

Regarding differences in sex composition: we agree that this is a key biological variable. Accordingly, we adjust for sex as a covariate in models that include both males and females to control bias. However, the genome-wide (mERRBs) analysis included females only because we had limited funds for assays and thus, sought to improve statistical power by minimizing the need for covariate adjustment. In response to this comment, we now mention our sampling design, the relevance of sex as a key biological variable, and limitations of the female-only sample for the mERRBS explicitly when describing covariates in the Supporting Material and in the discussion of limitations.

Supporting Information:

Pgs 12 (SI), Lines 227-228: In this species, females are higher ranking, more aggressive, and morphologically larger than males, providing greater access to resources for them and their dependent offspring²⁸.

Discussion of limitations:

Pg 39, Lines 749-755: Furthermore, aspects of the study design limit causal inference. In particular, our genome-wide analysis was limited by budgetary constraints so sample selection focused on animals of similar age and sex for the sake of statistical power. Consequently, results of this particular analysis may not be generalizable to both sexes and precludes analyses of between sex comparisons. We acknowledge the tradeoff between improved statistical power and internal

validity at the expense of external validity of results given the opportunistic nature of this analysis.

3. **Comment:** Furthermore, the usage of whole blood samples is tricky, blood-cell composition quickly changes e.g. in case of an infection numbers of leucocytes are increasing. The cell composition change is accompanied by a change in methylation. The authors cannot control for this – as stated in their response to reviewer 1.

Response: We acknowledge this limitation and have addressed it per the suggestion of Reviewer #1 (see our response to Rev #1 Comment #1).

4. **Comment:** Despite the authors' best effort, the results look fragile and dependent on the method used. I suggest applying a measure of sensitivity or even more holistic approach, such as a multiverse analysis proposed by Gelman (<https://doi.org/10.1177/1745691616658637>), to increase the study's significance.

Response: We agree with the reviewer regarding the importance of reproducibility. In accordance with Andrew Gelman's suggestions to compare the robustness of results across multiple data sets and models, we have indeed run multiple models on multiple datasets, and across our revisions, have now implemented multiple data processing and QA/QC pipelines. Across these multiple analyses, results have remained consistent. For instance:

- Part three of our analysis, described beginning on Pg 25 and summarized in SI Table 6, reproduces the models and, for the most part the results (i.e. the direction of estimates of association are unchanged) from previous parts 1 and 2 (Table 1, Figures 6-8) of our analysis while using a smaller subset of data for which there is complete covariate overlap.
- Across all results, we report estimates from both unadjusted and adjusted models to provide transparency of the stability of estimates (c.f. Table 1, Figures 6-8) after accounting for three major categories of covariates (demographic covariates; early social experience covariates; and ecological covariates) – both individually and all together in the same model. The consistency of the direction and magnitude of associations across multiple variable in the face of the moderate sample sizes models (e.g., Figure 6) support robustness of our findings.
- Finally, in the interest of reproducibility and transparency, we will submit R scripts alongside of our data and results, annotated with inclusion/exclusion decisions such that any interested reader can confirm the robustness of our findings across the multiple analyses we implemented.

5. **Comment:** While the idea and species' system is exciting, at this stage I am not convinced the results provide deeper knowledge into the species system.

Response: We are encouraged that this reviewer finds the study system exciting. Given the challenges of studying wild animals, and constraints of the present opportunistic study which leverages extant data and archived biospecimens, we reiterate that this multi-part analysis is opportunistic, one-of-a-kind, and represents a first step into understanding the relationship among the type and timing early social experiences, molecular mechanisms, and stress physiology in a wild animal. With additional follow up of these study animals and with funding obtained specifically to

collect additional data, we hope to be better equipped to refine and test new hypotheses in spotted hyenas. We have incorporated these sentiments into the conclusion section on pages 39-40, Lines 761-770.

REVIEWERS' COMMENTS

Reviewer #1 (Remarks to the Author):

I believe the authors have addressed my concerns to the greatest degree possible given the current dataset. I am still concerned about the possibility of cell type confounds, but find the authors' additions to the text satisfactory. I think this paper would benefit greatly from an enhanced genome-wide methylation dataset which is better connected to the global methylation results, but understand and am sympathetic to the opportunistic nature of sample collection (a point the authors make very clear in their response to Rev 4 comment 2). Hopefully sample collection in future years will help!

Reviewer #4 (Remarks to the Author):

Thanks to authors for their answers, their complementary information and especially their effort to make the manuscript more transparent.

Generally, I repeat my concerns about the robustness of the findings, - the "The garden of forking paths" problem (cf: Gelman et al. 2013, see also previous comments/answers), that I see as a serious limitation not directly addressed in the manuscript nor in the authors' answers. In other words, a gap remains between what the data can show and what they are claimed to show. In these cases, nuances are one way to render the complexity of a question.

My general comment can be exemplified by the sample material used, whole blood. Whole blood has different cell-types which easily and rapidly change in composition. Because different cell types have different methylation patterns, the composition of the blood cell-type is known to influence methylation pattern. In the current study, quantitative epigenetic data were generated from whole blood with unknown cell-type composition. These data are fragile in the sense that methylation patterns may vary widely based on the abundance of blood cell-types. These quantitative data were subsequently modeled with other parameters, passing the lower-level fragility to higher-level models. Inferences were finally drawn from the effect-sizes of some parameters of these models. The lack of a clear study design – which the authors call "opportunistic sampling", leads to various factors that cannot be controlled for. Thus the results carry an inherent fragility which is not problematic per se, but needs to be clearly pointed out in the manuscript.

Appreciatory, some of the comments raised during the review process have been made more transparent in the conclusion section. These additions may be moved to the discussion section, under a header (e.g. "Limitations of this study") to highlight them to other scientists in the field.

I hope my comments were helpful to increase the clarity of the manuscript's content, and to improve its quality. I leave it up to the editors to decide if the manuscript meets the standards of Nature Communications.

Point-by-point response to reviewers' comments

Reviewer #1

1. **Comment:** I believe the authors have addressed my concerns to the greatest degree possible given the current dataset. I am still concerned about the the possibility of cell type confounds, but find the authors additions to the text satisfactory. I think this paper would benefit greatly from an enhanced genome-wide methylation dataset which is better connected to the global methylation results, but understand and am sympathetic to the opportunistic nature of sample collection (a point the authors make very clear in their response to Rev 4 comment 2). Hopefully sample collection in future years will help!

Response: We agree with the reviewer's assessment and agree that an enhanced genome-wide DNA methylation dataset will be beneficial. We have included this in the Limitation of this study section.

Pg 26, Lines 418-421:

Further analyses that aim to test hypotheses about functional genomic pathways involving a larger sample of genome-wide DNA methylation data are warranted. With larger sample sizes and additional annotation of the hyena genome, future studies could home in on specific genomic contexts containing differentially methylated CpG sites and identify pathways enriched in response to early-life social experience.

Reviewer #4

1. **Comment:** Thanks to authors for their answers, their complementary information and especially their effort to make the manuscript more transparent. Generally, I repeat my concerns about the robustness of the findings, - the "The garden of forking paths" problem (cf: Gelman et al. 2013, see also previous comments/answers), that I see as a serious limitation not directly addressed in the manuscript nor in the authors` answers. In other words, a gap remains between what the data can show and what they are claimed to show. In these cases, nuances are one way to render the complexity of a question. My general comment can be exemplified by the sample material used, whole blood. Whole blood has different cell-types which easily and rapidly change in composition. Because different cell types have different methylation patterns, the composition of the blood cell-type is known to influence methylation pattern. In the current study, quantitative epigenetic data were generated from whole blood with unknown cell-type composition. These data are fragile in the sense that methylation patterns may vary widely based on the abundance of blood cell-types. These quantitative data were subsequently modeled with other parameters, passing the lower-level fragility to higher-level models. Inferences were finally drawn from the effect-sizes of some parameters of these models. The lack of a clear study design – which the authors call "opportunistic sampling", leads to various factors that cannot be controlled for. Thus the results carry an inherent fragility which is not problematic per se, but needs to be clearly pointed out in the manuscript. Appreciatory, some of the comments raised during the review process have been made more transparent in the conclusion section. These additions may be

moved to the discussion section, under a header (e.g. “Limitations of this study”) to highlight them to other scientists in the field. I hope my comments were helpful to increase the clarity of the manuscript’s content, and to improve its quality. I leave it up to the editors to decide if the manuscript meets the standards of Nature Communications.

Response: We agree with the reviewer that this study is not without limitations. To address the above concerns, we have done the following:

- Addressed the issue of confounding by cell type, which was an issue raised by Reviewer #1 in the prior round of revisions. This reviewer had specifically suggested that we include the sentence below:

Pg 26, Lines 462-465:

Fourth, we used archived DNA from whole blood that lacked information on cell type composition, and therefore we were not able to control for cell type heterogeneity as a source of variation in DNA methylation. However, cell type composition may be influenced by factors like social stress¹⁰², and therefore might be on the causal pathway between the explanatory variables of interest and DNA methylation.

- As suggested, we now include a separate section for study limitations (page 25 starting at line 444), including the fact that we capitalized on available data and biospecimens (as opposed to starting an experiment from scratch), which is a strength in addition to being a limitation given that leveraging existing infrastructure and resources in a pre-existing wild animal population is, in many ways, a more efficient and robust approach to testing hypotheses than starting an entirely new study.
- Finally, to generally address the reviewer’s comment about our results and inference, we urge readers to interpret our findings cautiously throughout and make recommendations to improve upon the present study in the future. As a few examples:

Page 20, Lines 327-330:

Considering not only the discrepancies in how early-life experience and stress outcomes are measured, but also differences between controlled laboratory settings vs. studies of wild animals, we urge caution when comparing our results to previous studies involving captive primates and rodents.

Page 25, Lines 445-450:

First, we used a large database of existing behavioral, demographic, and biological variables to test our hypotheses. As a result, we lacked complete data overlap between different variables of interest resulting in somewhat differing sample sizes across analyses (thereby hindering our ability to make direct comparisons across analytical subsamples). Further, the observational nature of the data limit causal inference due to potential for unmeasured confounding.